# ENHANCING ZEROTH-ORDER FINE-TUNING FOR LANGUAGE MODELS WITH LOW-RANK STRUCTURES

**Yiming Chen**[†,*] **Yuan Zhang**[†,*] **Liyuan Cao**[‡] **Kun Yuan**[†,§,⌘] **Zaiwen Wen**[†]
[†]Peking University, Beijing, China;  [‡]Nanjing University, Nanjing, China
[§]AI for Science Institute, Beijing, China
[*]Equal contributions  [⌘]Corresponding author: `kunyuan@pku.edu.cn`

## ABSTRACT

Parameter-efficient fine-tuning (PEFT) significantly reduces memory costs when adapting large language models (LLMs) for downstream applications. However, traditional first-order (FO) fine-tuning algorithms incur substantial memory overhead due to the need to store activation values for back-propagation during gradient computation, particularly in long-context fine-tuning tasks. Zeroth-order (ZO) algorithms offer a promising alternative by approximating gradients using finite differences of function values, thus eliminating the need for activation storage. Nevertheless, existing ZO methods struggle to capture the low-rank gradient structure common in LLM fine-tuning, leading to suboptimal performance. This paper proposes a low-rank ZO gradient estimator and introduces a novel **lo**w-rank **ZO** algorithm (LOZO) that effectively captures this structure in LLMs. We provide convergence guarantees for LOZO by framing it as a subspace optimization method. Additionally, its low-rank nature enables LOZO to integrate with momentum techniques while incurring negligible extra memory costs. Extensive experiments across various model sizes and downstream tasks demonstrate that LOZO and its momentum-based variant outperform existing ZO methods and closely approach the performance of FO algorithms.

## 1 INTRODUCTION

Large language models (LLMs) have demonstrated exceptional performance across a wide range of domains (Solaiman et al., 2019; Brown, 2020; Achiam et al., 2023). To adapt LLMs for specific downstream applications, fine-tuning pre-trained models has become the *de facto* approach (Gururangan et al., 2020; Sanh et al., 2021). Parameter-efficient fine-tuning (PEFT) methods, such as those proposed by (Hu et al., 2021; Lester et al., 2021), aim to reduce memory consumption by freezing most pre-trained weights and updating only a subset of parameters. However, even with these approaches, first-order (FO) optimization algorithms like stochastic gradient descent (SGD) (Amari, 1993) and Adam (Kingma, 2014) still encounter substantial memory overhead due to the need to store activation values for back-propagation during gradient computation. This issue becomes even more pronounced in long-context settings where activations dominate memory usage.

To enhance memory efficiency, a promising alternative is the use of zeroth-order (ZO) algorithms (Spall, 1992). Unlike FO methods, ZO algorithms do not require direct gradient computation. Instead, they approximate gradients using finite differences of function values, eliminating the need for back-propagation and the storage of activation values, which leads to substantial memory savings. ZO algorithms have been extensively studied over the past few decades (Duchi et al., 2015; Nesterov & Spokoiny, 2017; Berahas et al., 2022) and were recently applied to fine-tuning LLMs for the first time in (Malladi et al., 2023), where the authors adapt the classical ZO stochastic gradient descent (ZO-SGD) algorithm (Ghadimi & Lan, 2013) to a memory-efficient variant known as the MeZO algorithm. As demonstrated in (Malladi et al., 2023), the MeZO algorithm can reduce memory costs to a quarter of those incurred by standard SGD.

However, ZO algorithms still face several challenges when fine-tuning LLMs. A primary concern is the substantial discrepancy in the matrix rank between estimated ZO gradients and true FO gradients. Extensive literature reports that FO gradients generated during backpropagation in LLM fine-tuning

exhibit a *low-rank* structure (Malladi et al., 2023; Zhao et al., 2024a; Hao et al., 2024). In contrast, the ZO gradients in MeZO are derived from a perturbation matrix sampled from a Gaussian distribution, which is nearly *full-rank*. This discrepancy can result in a performance gap between ZO and FO algorithms. Another challenge is constructing momentum algorithms using ZO gradients. Since ZO gradients are typically full-rank, the resulting momentum variable also becomes full-rank, leading to significant overhead in storing the momentum (Malladi et al., 2023; Zhang et al., 2024; Chen et al., 2019). This overhead offsets the memory efficiency gained through ZO algorithms.

It is evident that the above challenges arise from the full-rank structure of ZO gradient estimates. To address these issues, this work presents a novel strategy for the gradient approximation based on finite differences of function values. Specifically, we propose a **l**ow-rank matrix-wise **g**radient **e**stimator (LGE) scheme and a new variant of the ZO-SGD algorithm that capitalizes on this approach. This ensures that our ZO gradient estimator consistently retains a low-rank structure, fluctuating within a low-dimensional subspace across iterations. As a result, our estimator more accurately approximates the low-rank FO gradient employed in LLM fine-tuning, leading to improved empirical performance. Moreover, the low-rank ZO gradient facilitates the use of low-rank momentum variables, significantly reducing memory usage compared to conventional ZO momentum algorithms. Our contributions can be summarized as follows:

- We propose a low-rank ZO gradient estimator. Unlike traditional coordinate-wise or randomized approaches, our approach derives the ZO gradient using a low-rank perturbation matrix, ensuring that the approximated gradient retains a low-rank structure. Our derived ZO gradient closely resembles the FO gradient structure in LLM fine-tuning.

- We develop two novel low-rank ZO algorithms for LLM fine-tuning: **Lo**w-rank **ZO**-SGD (LOZO) and its **m**omentum-based variant LOZO-M. A critical component in these algorithms is the *lazy sampling strategy*, where the low-rank random perturbation matrix is sampled over multiple steps, rather than at each iteration. This allows the ZO algorithm to sufficiently explore a low-rank subspace over a longer period, preventing per-iteration abrupt changes to model parameters and enhancing fine-tuning performance. Moreover, LOZO-M incurs negligible additional memory overhead for storing momentum.

- We establish convergence guarantees for LOZO under common assumptions in stochastic ZO optimization. A key insight from our convergence analysis is that LOZO can be viewed as a subspace optimization method employing a standard ZO gradient estimator. This method iteratively solve the fine-tuning problem by alternating between different low-rank subspaces to progressively improve the overall solution.

- We conduct extensive experiments across various model scales (ranging from 350M to 66B) and downstream tasks, including classification, multiple-choice, and generation. LOZO and LOZO-M outperform zero-shot, ICL, MeZO, and MeZO-LoRA in most tasks, while maintaining the same storage overhead as MeZO.

## 1.1 RELATED WORK

**Zeroth-order optimization.** Zeroth-order (ZO) optimization typically employs finite difference of function values to approximate gradients. Since it does not require gradient computation, ZO methods have been widely applied in various machine learning (ML) domains, including adversarial attack and defense (Ilyas et al., 2018; Zhao et al., 2019; Tu et al., 2019; Zhang et al., 2022b), model-agnostic contrastive explanations (Dhurandhar et al., 2019), and AutoML (Wang et al., 2022). ZO algorithms have been derived from FO methods in numerous studies, such as ZO-SGD (Ghadimi & Lan, 2013; Liu et al., 2019a), ZO-Adam (Chen et al., 2019), and ZO-SVRG (Liu et al., 2018; Ji et al., 2019), among others (Lian et al., 2016; Liu et al., 2020). Although straightforward, these adaptations often exhibit high variance and slow convergence because of the dimensionality of the model. To address this, techniques such as sparse gradient exploitation (Balasubramanian & Ghadimi, 2018; Cai et al., 2022) and feature reuse in deep neural networks (Chen et al., 2023) have been proposed, highlighting the potential of ZO optimization in large-scale ML problems.

**Memory-efficient fine-tuning**. A range of memory-efficient methods have been proposed to enhance the accessibility of LLM fine-tuning. For instance, LoRA (Hu et al., 2021) introduces low-rank perturbations to pre-trained model weights, utilizing only a few trainable parameters while achieving performance comparable to full fine-tuning. Other approaches (Zhao et al., 2024a; Hao

et al., 2024; Muhamed et al., 2024) compress gradients by projecting them into subspaces, thereby reducing the memory required for storing optimizer states. In contrast to first-order (FO) algorithms, zeroth-order (ZO) algorithms (Malladi et al., 2023) have gained considerable attention for their efficiency, as they avoid storing activation values. To accelerate ZO fine-tuning, Gautam et al. (2024) introduced ZO-SVRG for LLM fine-tuning, while Zhao et al. (2024b) employed ZO methods to approximate a natural gradient algorithm. Although these approaches improve convergence, they often incur increased memory costs. To address this trade-off, Liu et al. (2024) proposed incorporating a sparse mask into ZO iterations to reduce dimensionality and expedite fine-tuning, albeit at the cost of some accuracy. Li et al. (2024) explored a hybrid approach, combining FO gradients with ZO estimators in each iteration. Complementing these efforts, Zhang et al. (2024) provide a comprehensive benchmark of various ZO-based algorithms for LLM fine-tuning.

## 2 PRELIMINARIES

This section provides an overview of ZO optimization and the commonly used ZO gradient estimators. We also introduce the MeZO algorithm (Malladi et al., 2023) used in LLM fine-tuning.

### 2.1 ZEROTH-ORDER (ZO) OPTIMIZATION

We consider the following optimization problem:

$$\min_{\boldsymbol{X}} f(\boldsymbol{X}) := \mathbb{E}_{\xi}[F(\boldsymbol{X}; \xi)], \tag{1}$$

where $\boldsymbol{X}$ represents the set of trainable parameters with dimension $d$. For example, in the LLM fine-tuning process, we can express $\boldsymbol{X} = \{X_\ell\}_{\ell=1}^{\mathcal{L}}$, where $X_\ell \in \mathbb{R}^{m_\ell \times n_\ell}$ denotes the $\ell$-th weight matrix and $\mathcal{L}$ is the number of layers. The function $F(\boldsymbol{X}; \xi)$ is the loss function that depends on a random variable $\xi$.

The ZO method estimates gradients using only function evaluations, without requiring direct access to gradient information. Two commonly employed gradient estimation schemes are the deterministic Coordinate-wise Gradient Estimation (CGE) (Lian et al., 2016; Chen et al., 2023) and the Randomized vector-wise Gradient Estimation (RGE) (Spall, 1992; Duchi et al., 2015; Nesterov & Spokoiny, 2017). These are formally defined as:

$$(\text{CGE}) \quad \hat{\nabla}F(\boldsymbol{X}; \xi) := \sum_{i=1}^{d} \frac{F(\boldsymbol{X} + \epsilon \boldsymbol{E}_i; \xi) - F(\boldsymbol{X} - \epsilon \boldsymbol{E}_i; \xi)}{2\epsilon} \boldsymbol{E}_i, \tag{2}$$

$$(\text{RGE}) \quad \hat{\nabla}F(\boldsymbol{X}; \xi) := \frac{F(\boldsymbol{X} + \epsilon \boldsymbol{Z}; \xi) - F(\boldsymbol{X} - \epsilon \boldsymbol{Z}; \xi)}{2\epsilon} \boldsymbol{Z}. \tag{3}$$

The scalar $\epsilon$ denotes the perturbation magnitude, which influences the accuracy of the gradient approximation. Both $\boldsymbol{E}_i$ and $\boldsymbol{Z}$ are of the same dimensions as $\boldsymbol{X}$. The quantity $\boldsymbol{E}_i$ is a basis vector/matrix with its $i$-th element set to one and all other elements set to zero, whereas the elements of $\boldsymbol{Z}$ are randomly generated, typically sampled independently from a standard normal distribution. An extension of the RGE method is the $q$-RGE approach. Here, the RGE is computed $q$ times independently, and the final gradient estimate is obtained by averaging these estimations.

Utilizing these gradient estimators, the ZO-SGD method is implemented through the following iterative scheme:

$$\boldsymbol{X}^{t+1} = \boldsymbol{X}^t - \alpha \hat{\nabla}F(\boldsymbol{X}^t; \xi^t), \tag{4}$$

where $\alpha$ denotes the step size, also termed the learning rate, and $\hat{\nabla}F$ represents the gradient estimated with ZO information.

### 2.2 MEMORY-EFFICIENT ZO-SGD (MEZO)

The standard implementation of ZO-SGD incurs substantial memory costs. For example, when constructing the gradient estimator using the RGE scheme, the traditional ZO-SGD method requires memory to store the perturbation matrix $\boldsymbol{Z}$. To mitigate this memory overhead, the MeZO method (Malladi et al., 2023) was introduced as a memory-efficient variant of ZO-SGD. Unlike the standard approach, MeZO avoids storing the entire perturbation matrix $\boldsymbol{Z}$. Instead, the algorithm performs

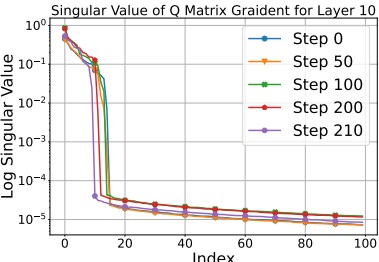 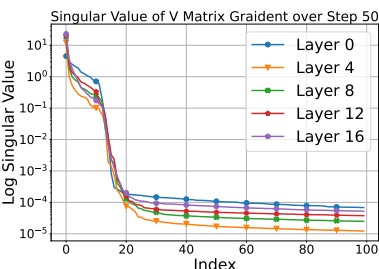

Figure 1: The low-rank structure of the gradients encountered in the fine-tuning of LLMs, demonstrated using the OPT-1.3B model with the COPA dataset, where the gradient matrices have dimensions of $2048 \times 2048$. For both two figures, we report only the 100 largest singular values. *Left*: Singular value distribution of the gradient of the attention $Q$ matrix in layer 10 across different training steps. *Right*: Singular value distribution of the gradient of the attention $V$ matrix across different layers at training step 50.

both the perturbation and ZO-SGD updates in place and employs a technique of saving the random seed used to generate $\boldsymbol{Z}$, allowing it to be regenerated when necessary. While this introduces additional computational costs, it significantly reduces memory usage.

## 3   LOW-RANK ZEROTH-ORDER GRADIENT ESTIMATOR

**LLM gradients exhibit low-rank structures.** The low-rank structure of neural networks has been extensively investigated in previous literature (Li et al., 2018; Larsen et al., 2021). These studies suggest that loss landscapes exist within an *intrinsic dimension*, implying that model weights can be optimized within a low-rank subspace. Furthermore, additional researches (Sagun et al., 2017; Gur-Ari et al., 2018) have demonstrated that stochastic gradients dynamically converge to a remarkably small subspace, especially when fine-tuning LLMs (Zhang et al., 2023). Recent work (Zhao et al., 2024a) also provides theoretical evidence that the gradient matrix becomes low-rank during LLM training and fine-tuning. Figure 1 investigates the rank of the gradient matrix during LLM fine-tuning. The left plot demonstrates that the gradient matrix remains low-rank across training steps, while the right plot shows that this low-rank property persists across different layers.

**Existing ZO methods are ineffective at capturing low-rank structures.** Commonly used ZO gradient estimators like CGE and RGE cannot capture the low-rank structure of the true gradients during LLM fine-tuning. The RGE scheme produces an estimated gradient that is essentially a projection of the true gradient onto a standard normal random matrix $\boldsymbol{Z}$, which is almost always full rank. On the other hand, although the CGE scheme can approximate the full true gradient, it requires evaluating the loss function $d$ times per iteration, making it impractical for large-scale problems. As a result, existing ZO algorithms (Ghadimi & Lan, 2013; Liu et al., 2019a; Chen et al., 2019; Liu et al., 2018; Ji et al., 2019; Balasubramanian & Ghadimi, 2018; Cai et al., 2022; Chen et al., 2023; Malladi et al., 2023; Zhang et al., 2024), which rely on either CGE or RGE, fail to effectively account for the low-rank structure inherent in the gradient matrix during LLM fine-tuning.

**Low-rank ZO gradient estimator (LGE).** To bridge this gap, we propose a matrix-wise ZO gradient estimator, LGE, that preserves the low-rank structure in gradients. In LLM fine-tuning, let $\boldsymbol{X} = \{X_\ell\}_{\ell=1}^{\mathcal{L}}$ represent the model's weights, where $X_\ell \in \mathbb{R}^{m_\ell \times n_\ell}$ is the weight matrix of the $\ell$-th layer. We sample two matrices, $U_\ell \in \mathbb{R}^{m_\ell \times r_\ell}$ and $V_\ell \in \mathbb{R}^{n_\ell \times r_\ell}$, from standard normal distributions, where $r_\ell \ll \min\{m_\ell, n_\ell\}$. The LGE for partial gradient of the $\ell$-th weight matrix is defined as

$$\hat{\nabla}_{X_\ell} F(\boldsymbol{X}; \xi) := \frac{F(\{X_\ell + \epsilon U_\ell V_\ell^T\}_{\ell=1}^{\mathcal{L}}; \xi) - F(\{X_\ell - \epsilon U_\ell V_\ell^T\}_{\ell=1}^{\mathcal{L}}; \xi)}{2\epsilon} (U_\ell V_\ell^T / r_\ell). \quad (5)$$

The scaling factor $1/r_\ell$ is introduced to ensure that LGE is an unbiased estimator of the true gradient as $\epsilon \to 0$ (see Proposition A.1). Defining $\boldsymbol{U} := \{U_\ell\}_{\ell=1}^{\mathcal{L}}$, $\boldsymbol{V} := \{V_\ell\}_{\ell=1}^{\mathcal{L}}$, $\boldsymbol{r} := \{r_\ell\}_{\ell=1}^{\mathcal{L}}$, and $\hat{\nabla} F(\boldsymbol{X}; \xi) := \{\hat{\nabla}_{X_\ell} F(\boldsymbol{X}; \xi)\}_{\ell=1}^{\mathcal{L}}$, we express $\boldsymbol{X} \pm \epsilon \boldsymbol{U} \boldsymbol{V}^T := \{X_\ell \pm \epsilon U_\ell V_\ell^T\}_{\ell=1}^{\mathcal{L}}$ and $\boldsymbol{U} \boldsymbol{V}^T / \boldsymbol{r} := \{U_\ell V_\ell^T / r_\ell\}_{\ell=1}^{\mathcal{L}}$. Using these notations, LGE can be written into a more compact form:

$$\text{(LGE)} \qquad \hat{\nabla} F(\boldsymbol{X}; \xi) := \frac{F(\boldsymbol{X} + \epsilon \boldsymbol{U} \boldsymbol{V}^T; \xi) - F(\boldsymbol{X} - \epsilon \boldsymbol{U} \boldsymbol{V}^T; \xi)}{2\epsilon} (\boldsymbol{U} \boldsymbol{V}^T / \boldsymbol{r}). \quad (6)$$

Using the definition in (5), we observe that the gradient matrix $\hat{\nabla}_{X_\ell} F(\boldsymbol{X}; \xi)$ has a rank of at most $r_\ell$, effectively capturing the low-rank structure of the FO gradient in LLM fine-tuning.

# 4 LOW-RANK ZEROTH-ORDER SGD

This section introduces LOZO, a novel low-rank ZO algorithm for LLM fine-tuning. LOZO can be interpreted as a ZO subspace optimization method that leverages a standard gradient estimator. Based on this key insight, we establish convergence guarantees for LOZO.

## 4.1 ALGORITHM DEVELOPMENT

**Vanilla recursion.** Following the LGE definition (6), we introduce the LGE operator

$$\text{LGE}(\boldsymbol{X}, \boldsymbol{U}, \boldsymbol{V}, \boldsymbol{r}, \epsilon, \xi) := \frac{F(\boldsymbol{X} + \epsilon \boldsymbol{U}\boldsymbol{V}^T; \xi) - F(\boldsymbol{X} - \epsilon \boldsymbol{U}\boldsymbol{V}^T; \xi)}{2\epsilon} (\boldsymbol{U}\boldsymbol{V}^T/\boldsymbol{r}). \tag{7}$$

To solve problem (1), the vanilla recursion with the LGE scheme is as follows. For any $t \geq 0$,

$$\boldsymbol{X}^{t+1} = \boldsymbol{X}^t - \alpha \hat{\nabla} F(\boldsymbol{X}^t; \xi^t) \quad \text{where} \quad \hat{\nabla} F(\boldsymbol{X}^t; \xi^t) = \text{LGE}(\boldsymbol{X}^t, \boldsymbol{U}^t, \boldsymbol{V}^t, \boldsymbol{r}, \epsilon, \xi^t). \tag{8}$$

In practice, we only need to store $U_\ell$ and $V_\ell$ for each layer $\ell$, and we apply the perturbation (6) and the update (8) layer by layer, eliminating the need to retain the full gradient estimator $U_\ell V_\ell^T$. Since $r_\ell \ll \min\{m_\ell, n_\ell\}$, the additional memory required for storing $U_\ell$ and $V_\ell$ is negligible. Moreover, memory costs can be further reduced using the random seed technique (Malladi et al., 2023). Instead of storing $U_\ell$ and $V_\ell$ directly, only the random seeds $s_\ell^U$ and $s_\ell^V$ used to generate them are saved. Whenever $U_\ell$ and $V_\ell$ are needed, the seeds $s_\ell^U$ and $s_\ell^V$ are used to regenerate these matrices, thereby eliminating the need for their storage. While this approach reduces memory usage, it introduces additional floating-point operations (flops) due to the regeneration process.

**Lazy sampling strategy.** In the main recursion (8), the variable $\boldsymbol{X}^t$ is updated within the subspace spanned by $\boldsymbol{U}^t$ and $\boldsymbol{V}^t$ at each iteration $t$. However, if $\boldsymbol{U}^t$ and $\boldsymbol{V}^t$ are resampled at every iteration, the subspace will shift too frequently. This limits the algorithm's ability to adequately explore one low-rank subspace over a longer period, potentially causing abrupt changes in the model parameters $\boldsymbol{X}$ at each iteration and degrading fine-tuning performance.

Additionally, ZO methods capture less information about the true gradient compared to FO algorithms, necessitating more iterations to achieve similar performance. In other words, multiple ZO steps may be required to match the progress of a single FO step. This suggests that maintaining a low-rank structure in the gradient estimator at each step is insufficient; instead, the cumulative sum of gradient estimators over several consecutive iterations must also preserve a low-rank structure.

The motivations outlined above lead us to propose a lazy sampling strategy. While $\boldsymbol{U}$ is sampled at every iteration $t$, we only sample $\boldsymbol{V}$ every $\nu$ iterations, where $\nu > 0$ represents the chosen period duration. During the iterations $t \in \{k\nu, \ldots, (k+1)\nu - 1\}$ for each period $k$, the matrix $\boldsymbol{V}^{(k)}$ remains fixed, thus restricting the model update to the subspace spanned by $\boldsymbol{V}^{(k)}$. This leads to our proposed LOZO algorithm, whose update rule for any $t \geq 0$ is defined as:

$$\boldsymbol{X}^{t+1} = \boldsymbol{X}^t - \alpha \hat{\nabla} F(\boldsymbol{X}^t; \xi^t), \quad \text{where} \quad \hat{\nabla} F(\boldsymbol{X}^t; \xi^t) = \text{LGE}(\boldsymbol{X}^t, \boldsymbol{U}^t, \boldsymbol{V}^{(k)}, \boldsymbol{r}, \epsilon, \xi^t). \tag{9}$$

With the lazy sampling strategy, $\hat{\nabla} F(\boldsymbol{X}^t; \xi^t)$ consistently lies within the subspace determined by $\boldsymbol{V}^{(k)}$ for any $t \in \{k\nu, \ldots, (k+1)\nu - 1\}$. Therefore, the accumulation of the estimated gradients over these consecutive $\nu$ steps, which can be viewed as a more accurate approximation of the true gradient in a single FO step, has a rank that does not exceed $\boldsymbol{r}$. When $\nu = 1$, the LOZO update rule (9) reduces to the standard recursion (8). LOZO (9) can be implemented in Algorithm 1 with memory efficiency.

**Hyperparameter tuning.** The parameter $\nu$, which defines the number of steps over which $\boldsymbol{X}^t$ is updated within the same subspace, is critical for performance and should be set to a moderate value. If $\nu$ is too small, frequent subspace shifts may lead to abrupt model changes, while a $\nu$ that is too large limits the algorithm to focus on only a few subspaces, potentially reducing generalization. The parameter $\boldsymbol{r}$ defines the rank of the gradient estimator. Since the true gradient rank is unknown, we typically set $\boldsymbol{r}$ as a small constant that is significantly less than both $m_\ell$ and $n_\ell$ to avoid additional memory overhead. In our experiments, we set $r_\ell = r$ through all layers. The typical choices for the parameters are $r = 2, 4, 8$ and $\nu = 50, 100$.

---

**Algorithm 1:** Low-rank ZO-SGD (LOZO)

---

**Input:** parameters $\boldsymbol{X}$, loss function $F(\boldsymbol{X};\xi)$, step budget $T$, perturbation scale $\epsilon$, learning rate
    $\alpha$, sample interval $\nu$ and rank $\{r_\ell\}$.

**for** $t = 0, \ldots, T-1$ **do**
  **foreach** $X_\ell \in \boldsymbol{X}$ **do**
    Sample $U_\ell \in \mathbb{R}^{m_\ell \times r_\ell}$ from the standard normal distribution ;
    **if** $t \bmod \nu = 0$ **then**
      Sample $V_\ell \in \mathbb{R}^{n_\ell \times r_\ell}$ from the standard normal distribution ;  // Resample $V_\ell$

  $\boldsymbol{X} \leftarrow \text{Perturbation}(\boldsymbol{X}, \epsilon, \{U_\ell, V_\ell\})$ ;
  $F_+ \leftarrow F(\boldsymbol{X}; \xi)$ ;
  $\boldsymbol{X} \leftarrow \text{Perturbation}(\boldsymbol{X}, -2\epsilon, \{U_\ell, V_\ell\})$ ;
  $F_- \leftarrow F(\boldsymbol{X}; \xi)$ ;
  $\boldsymbol{X} \leftarrow \text{Perturbation}(\boldsymbol{X}, \epsilon, \{U_\ell, V_\ell\})$;           // Reset parameters
  $c \leftarrow (F_+ - F_-)/2\epsilon$ ;           // Calculate finite difference
  **foreach** $X_l \in \boldsymbol{X}$ **do**
    $X_\ell \leftarrow X_\ell - \alpha \cdot c(U_\ell V_\ell^T / r_l)$ ;     // Update parameters in place

**Function** $\text{Perturbation}(\boldsymbol{X}, \epsilon, \{U_\ell, V_\ell\})$:
  **foreach** $X_\ell \in \boldsymbol{X}$ **do**
    $X_\ell \leftarrow X_\ell + \epsilon U_\ell V_\ell^T$ ;        // Modify parameters in place
  **return** $\boldsymbol{X}$ ;

---

## 4.2 LOZO IS ESSENTIALLY A ZEROTH-ORDER SUBSPACE OPTIMIZATION METHOD

This section offers an in-depth understanding on LOZO, showing that it can be interpreted as a ZO subspace optimization method. This interpretation provides an insight into why LOZO can effectively solve problem (1) even with low-rank gradient estimates and a lazy sampling strategy.

**Random coordinate minimization.** We begin by revisiting the classical coordinate minimization algorithm for high-dimensional optimization problems. Consider the unconstrained minimization problem $\min_{x \in \mathbb{R}^d}\{f(x)\}$, where the dimension $d$ is exceedingly large. A commonly used and efficient method to tackle this problem is the coordinate minimization algorithm. Let $I := [e_1; \cdots; e_d]$ be the $d$-dimensional identity matrix, where $e_i$ represents the $i$-th basis vector. The random coordinate minimization algorithm then iterates as follows:

$$b_k^\star = \arg\min_{b \in \mathbb{R}}\{f(x^k + b \cdot e_{i_k})\}, \text{ where } i_k \sim \mathcal{U}\{1, \ldots, d\}, \tag{10}$$

$$x^{k+1} = x^k + b_k^\star \cdot e_{i_k}, \tag{11}$$

where $x^0$ is the initialized variable. The coordinate minimization algorithm has strong convergence guarantees, as demonstrated in studies such as (Hong et al., 2017; Tseng, 2001; Wright, 2015).

**Random subspace minimization.** Analogous to random coordinate minimization, the random subspace minimization approach is to solve optimization problems involving large matrix variables. To solve problem (1), i.e., $\min_{\boldsymbol{X}} f(\boldsymbol{X}) = \mathbb{E}_\xi[F(\boldsymbol{X}; \xi)]$, we can employ the following recursions

$$\boldsymbol{B}_k^\star = \arg\min_{\boldsymbol{B}}\{f(\boldsymbol{X}^{(k)} + \boldsymbol{B}(\boldsymbol{V}^{(k)})^T)\}, \text{ where } \boldsymbol{V}^{(k)} \text{ follows a normal distribution,} \tag{12}$$

$$\boldsymbol{X}^{(k+1)} = \boldsymbol{X}^{(k)} + \boldsymbol{B}_k^\star (\boldsymbol{V}^{(k)})^T. \tag{13}$$

Here, $\boldsymbol{V}$ is a low-rank matrix and $\boldsymbol{B}$ is a matrix variable with much smaller size than $\boldsymbol{X}$. For example, in the $\ell$-th layer, the matrix variable $B_\ell$ has dimensions $m_\ell \times r_\ell$, while $X_\ell$ has dimensions $m_\ell \times n_\ell$. The random subspace optimization approach addresses the original problem (1) by solving a series of subproblems that involve significantly smaller matrix variables, as shown in (12).

**ZO subspace optimization method.** To solve the $k$-th subproblem in (12), we apply the standard ZO-SGD and iterate for $\nu$ steps. The result is then used as an inexact solution to (12), followed by the update step in (13). Specifically, the update rule for the ZO subspace method is given by:

$$\boldsymbol{B}^{(k,s+1)} = \boldsymbol{B}^{(k,s)} - \boldsymbol{\gamma}\hat{\nabla}_{\boldsymbol{B}} F(\tilde{\boldsymbol{X}}^{(k)} + \boldsymbol{B}^{(k,s)}(\boldsymbol{V}^{(k)})^T; \xi^{(k,s)}), \quad s = 0, \cdots, \nu - 1, \tag{14a}$$

$$\tilde{\boldsymbol{X}}^{(k+1)} = \tilde{\boldsymbol{X}}^{(k)} + \boldsymbol{B}^{(k,\nu)}(\boldsymbol{V}^{(k)})^T. \tag{14b}$$

Here, we initialize $\boldsymbol{B}^{(k,0)} = \boldsymbol{0}$, the superscript $k$ indicates that the recursion is solving the $k$-th subproblem in (12), $\boldsymbol{\gamma}$ is the step size, and $\hat{\nabla}_{\boldsymbol{B}} F(\tilde{\boldsymbol{X}}^{(k)} + \boldsymbol{B}^{(k,s)}(\boldsymbol{V}^{(k)})^T; \xi^{(k,s)})$ is computed using the RGE scheme. Specifically, we sample $\boldsymbol{U}^{(k,s)}$ from a normal distribution and compute:

$$\hat{\nabla}_{\boldsymbol{B}} F(\tilde{\boldsymbol{X}}^{(k)} + \boldsymbol{B}^{(k,s)}(\boldsymbol{V}^{(k)})^T; \xi^{(k,s)}) \tag{15}$$
$$= \frac{F(\tilde{\boldsymbol{X}}^{(k)}+(\boldsymbol{B}^{(k,s)}+\epsilon\boldsymbol{U}^{(k,s)})(\boldsymbol{V}^{(k)})^T; \xi^{(k,s)})-F(\tilde{\boldsymbol{X}}^{(k)}+(\boldsymbol{B}^{(k,s)}-\epsilon\boldsymbol{U}^{(k,s)})(\boldsymbol{V}^{(k)})^T; \xi^{(k,s)})}{2\epsilon}\boldsymbol{U}^{(k,s)}.$$

**Equivalence between ZO subspace method and LOZO algorithm.** The ZO subspace method (14) is equivalent to the LOZO algorithm (9) when the step size $\boldsymbol{\gamma} = \alpha/\boldsymbol{r}$, as will be shown below. If both algorithms are initialized identically, i.e., $\boldsymbol{X}^0 = \tilde{\boldsymbol{X}}^{(0)}$, by comparing the update rules for $\boldsymbol{X}^t$ in LOZO (9) and (6), and for $\boldsymbol{B}$ in the ZO subspace method (14a) and (15), it is straightforward to see that $\boldsymbol{X}^t = \tilde{\boldsymbol{X}}^{(k)} + \boldsymbol{B}^{(k,s)}(\boldsymbol{V}^{(k)})^T$ holds when $t = k\nu + s$ and $s \in \{0, 1, \dots, \nu\}$ (see the detailed derivation in Appendix A.2). Thus, we have $\boldsymbol{X}^{k\nu} = \tilde{\boldsymbol{X}}^{(k)}$ for any $k$, which implies the equivalence between these two methods.

## 4.3 CONVERGENCE ANALYSIS

Based on the aforementioned interpretation, we can establish convergence guarantees for the LOZO algorithm. We adopt the following assumptions, which are standard in stochastic optimization.

**Assumption 4.1.** *For any $\xi$, the function $F(\boldsymbol{X}; \xi)$ is differentiable with respect to $\boldsymbol{X}$. Furthermore,*

- *The gradient $\nabla F(\boldsymbol{X}; \xi)$ is uniformly $L$-Lipschitz continuous, i.e., $\forall \boldsymbol{X}, \boldsymbol{Y}$,*

$$\|\nabla F(\boldsymbol{X}; \xi) - \nabla F(\boldsymbol{Y}; \xi)\| \le L\|\boldsymbol{X} - \boldsymbol{Y}\|, \quad \forall \xi, \tag{16}$$

  *where $\|\boldsymbol{X}\| := \sqrt{\sum_{\ell=1}^{\mathcal{L}} \|X_\ell\|_F^2}$ for any $\boldsymbol{X} = \{X_\ell\}_{\ell=1}^{\mathcal{L}}$.*
- *The stochastic gradient is unbiased and has bounded variance, i.e., $\forall \boldsymbol{X}$,*

$$\mathbb{E}[\nabla F(\boldsymbol{X}; \xi)] = \nabla f(\boldsymbol{X}) \quad \text{and} \quad \mathbb{E}\|\nabla F(\boldsymbol{X}; \xi) - \nabla f(\boldsymbol{X})\|^2 \le \sigma^2. \tag{17}$$

**Assumption 4.2.** *The random matrix $\boldsymbol{V} = \{V_\ell\}_{\ell=1}^{\mathcal{L}}$ is drawn from a distribution such that $V_\ell^T V_\ell = n_\ell I$ and $\mathbb{E}[V_\ell V_\ell^T] = r_\ell I$ for each $\ell$.*

**Remark 4.3.** *Given that $n_\ell \gg r_\ell$ in practice, when each $V_\ell$ is drawn from a standard normal distribution, it typically holds that $V_\ell^T V_\ell \approx n_\ell I$. However, to rigorously satisfy Assumption 4.2, each $V_\ell$ should be drawn from a Haar distribution or as a random coordinate matrix (see (Kozak et al., 2023), Examples 1 and 2 for more details).*

The following theorem establishes the convergence rate of the LOZO algorithm, with the detailed proof provided in Appendix B.

**Theorem 4.4.** *Under Assumptions 4.1 and 4.2, and letting $T = K\nu$, with suitable choices of $\alpha$ and $\epsilon$, the sequence of the $k\nu$-th variables $\{\boldsymbol{X}^{k\nu}\}$ generated by LOZO converges at the following rate:*

$$\frac{1}{K} \sum_{k=0}^{K-1} \mathbb{E}\|\nabla f(\boldsymbol{X}^{k\nu})\|^2 \le O\left(\sqrt{\frac{\Delta_0 L\tilde{d}\sigma^2}{T}} + \frac{\Delta_0 Ld\nu}{T}\right), \tag{18}$$

*where $\Delta_0 := f(\boldsymbol{X}^0) - f^*$, $\tilde{d} = \sum_{\ell=1}^{\mathcal{L}}(m_\ell n_\ell^2/r_\ell)$ and $d = \sum_{\ell=1}^{\mathcal{L}} m_\ell n_\ell$.*

**Difference between LOZO and MeZO-LoRA.** Theorem 4.4 implies LOZO solves $\min_{\boldsymbol{X}} f(\boldsymbol{X}) := \mathbb{E}_\xi[F(\boldsymbol{X}; \xi)]$ directly even when using low-rank gradient estimates. In contrast, the MeZO-LoRA approach, which combines MeZO (Malladi et al., 2023) and LoRA (Hu et al., 2021), solves $\min_{\boldsymbol{A}, \boldsymbol{B}} f(\boldsymbol{X} + \boldsymbol{AB}) := \mathbb{E}_\xi[F(\boldsymbol{X} + \boldsymbol{AB}; \xi)]$. Notably, MeZO-LoRA can only optimize the low-rank adapters $\boldsymbol{A}$ and $\boldsymbol{B}$, without the capability to optimize the full parameter $\boldsymbol{X}$. This distinction explains why LOZO outperforms MeZO-LoRA in most of our empirical studies.

## 4.4 LOZO WITH MOMENTUM

**Momentum technique requires additional memory overhead.** The momentum technique is widely used in modern deep learning to reduce gradient variance and accelerate the convergence of training. The ZO-SGD with momentum (ZO-SGD-M) iteration can be expressed as follows:

$$\boldsymbol{M}^t = \beta \boldsymbol{M}^{t-1} + (1-\beta)\hat{\nabla}F(\boldsymbol{X}^t; \xi^t), \quad \boldsymbol{X}^{t+1} = \boldsymbol{X}^t - \alpha \boldsymbol{M}^t, \tag{19}$$

where $\boldsymbol{M}^t := \{M_l^t\}_{\ell=1}^{\mathcal{L}}$ is the momentum term, $\beta$ is the momentum coefficient, and $\hat{\nabla}F(\boldsymbol{X}^t; \xi^t)$ denotes the ZO estimated gradient at iteration $t$. Compared with the vanilla ZO-SGD, ZO-SGD-M introduces additional memory overhead since it requires storing the momentum term, which is proportional to the size of the model.

**The LOZO-M algorithm introduces negligible memory overhead.** We can integrate the proposed LOZO algorithm with the momentum technique (LOZO-M) without the memory overhead issue. To simplify the formulation, we denote the finite difference of function values at iteration $t$, where $t \in \{k\nu, \ldots, (k+1)\nu - 1\}$, as $c^t$. The gradient estimator for the LOZO algorithm, as introduced in (9), can then be represented as $\hat{\nabla}F(\boldsymbol{X}^t; \xi^t) = c^t \boldsymbol{U}^t (\boldsymbol{V}^{(k)}/\boldsymbol{r})^T$.

We now introduce LOZO-M, the memory-efficient version of ZO-SGD with momentum. The key observation is that $\boldsymbol{V}^{(k)}$ remains fixed for $t \in \{k\nu, \ldots, (k+1)\nu - 1\}$. Consequently, it suffices to accumulate $c^t \boldsymbol{U}^t$ instead of the full gradient estimator $\hat{\nabla}F(\boldsymbol{X}^t; \xi^t)$ for updating the momentum, thereby significantly reducing memory overhead. The update rule is expressed as follows:

$$\boldsymbol{N}^t = \beta \boldsymbol{N}^{t-1} + (1-\beta)c^t \boldsymbol{U}^t, \quad \boldsymbol{X}^{t+1} = \boldsymbol{X}^t - \alpha \boldsymbol{N}^t (\boldsymbol{V}^{(k)}/\boldsymbol{r})^T, \tag{20}$$

where $\boldsymbol{N}^t (\boldsymbol{V}^{(k)}/\boldsymbol{r})^T$ corresponds to the original momentum. When updating $\boldsymbol{X}^t$, we compute $\boldsymbol{N}^t (\boldsymbol{V}^{(k)}/\boldsymbol{r})^T$ layer by layer and discard the result immediately after updating the weights of each layer. Consequently, compared with the vanilla ZO-SGD-M (19), LOZO-M (20) only requires storing $\boldsymbol{N}^t := \{N_\ell^t\}_{\ell=1}^{\mathcal{L}}$. Since $r_\ell \ll \min(m_\ell, n_\ell)$, the additional memory overhead is negligible.

**Momentum projection.** When $\boldsymbol{V}^{(k)}$ is updated, an additional step is required. Specifically, at $t = (k+1)\nu$, the gradient estimator $\hat{\nabla}F(\boldsymbol{X}^t; \xi^t)$ takes the form $c^t \boldsymbol{U}^t (\boldsymbol{V}^{(k+1)}/\boldsymbol{r})^T$ due to resampling, while the momentum from the previous step is $\boldsymbol{N}^{t-1}(\boldsymbol{V}^{(k)}/\boldsymbol{r})^T$. Thus, we cannot update the momentum by simply combining $c^t \boldsymbol{U}^t$ and $\boldsymbol{N}^{t-1}$ as per (20). To address this, we project the old momentum onto the new subspace spanned by $\boldsymbol{V}^{(k+1)}$ before updating $\boldsymbol{N}^t$. We compute:

$$\tilde{\boldsymbol{N}}^{t-1} = \arg\min_{\boldsymbol{N}} \|\boldsymbol{N}^{t-1}(\boldsymbol{V}^{(k)})^T - \boldsymbol{N}(\boldsymbol{V}^{(k+1)})^T\|.$$

By doing this, the projected momentum $\tilde{\boldsymbol{N}}^{t-1}(\boldsymbol{V}^{(k+1)}/\boldsymbol{r})^T$ becomes a good approximation of the momentum from the previous step, taking the same form as the gradient estimator $\hat{\nabla}F(\boldsymbol{X}^t; \xi^t)$. Therefore, we use $\tilde{\boldsymbol{N}}^{t-1}$ to replace $\boldsymbol{N}^{t-1}$ in the momentum update (20) at $t = (k+1)\nu$. Given that $V_\ell^T V_\ell \approx n_\ell I$ for any $\ell$ (see Remark 4.3), the solution is $\tilde{\boldsymbol{N}}^{t-1} = \boldsymbol{N}^{t-1}(\boldsymbol{V}^{(k)})^T(\boldsymbol{V}^{(k+1)}/\boldsymbol{n})$, where $\boldsymbol{n} := \{n_\ell\}_{\ell=1}^{\mathcal{L}}$. The detailed LOZO-M algorithm is provided in Algorithm 2.

## 5 EXPERIMENTS

This section evaluates the performance of our algorithm across multiple tasks, including the Super-GLUE benchmark (Wang et al., 2019) and other datasets, as detailed in Appendix C.1. We compare our LOZO and LOZO-M algorithms with MeZO (Malladi et al., 2023) as well as its variants, and other baselines including zero-shot and in-context learning (ICL) approaches. We also test full fine-tuning and LoRA using the gradient-based Adam method (Kingma, 2014), referred to as FT and FT-LoRA, respectively.

Our experiments evaluate the algorithms on language models (LMs) of various scales, including RoBERTa-large (Liu et al., 2019b) and large autoregressive LMs such as OPT-13B, 30B, and 66B (Zhang et al., 2022a). We also test LLaMA models (Touvron et al., 2023) of varying scales, with the results presented in Appendix D.3. For a fair comparison, we conduct a full grid search of the parameters provided in (Malladi et al., 2023) and select the best results for MeZO and its variants.

## 5.1 MEDIUM-SIZED MASKED LANGUAGE MODELS

We conduct experiments employing RoBERTa-large on tasks including sentiment classification, natural language inference and topic classification. To mitigate the effects of random variability, all experimental results reported in this subsection are the means of the outcomes resulted from five different random seeds.

**LOZO outperforms MeZO and MeZO-LoRA on the majority of datasets and achieves performance comparable to FT.** As shown in Figure 2, our algorithm demonstrates a performance gain on the three listed datasets, with a particularly significant improvement on the MNLI dataset, surpassing both MeZO and MeZO-LoRA. The performance gap between LOZO and FT is generally less than 1%, and with a larger $k$, LOZO can further narrow this gap and even exceed its performance. We provide the complete results for RoBERTa-large in Appendix D.1.

**LOZO-M consumes similar memory usage to LOZO while delivering superior performance compared to MeZO and MeZO-Adam.** We measure the actual memory consumption during the training of RoBERTa-large using LOZO, MeZO, and their respective variants to evaluate the memory efficiency of our methods. As shown in Table 1, the additional memory overhead introduced by the momentum technique in LOZO is negligible. In contrast, MeZO fails to achieve this efficiency.

| Algorithm | MNLI | | SNLI | |
|---|---|---|---|---|
| | Accuracy (%) | Memory Usage (GB) | Accuracy (%) | Memory Usage (GB) |
| LOZO | 61.6 | 2.83 | 73.4 | 2.83 |
| LOZO-M | **62.7** | 2.84 | **74.0** | 2.84 |
| MeZO | 56.7 | 3.00 | 68.5 | 3.00 |
| MeZO-M | 58.9 | 5.89 | 69.6 | 5.89 |
| MeZO-Adam | 62.6 | 7.42 | 72.7 | 7.42 |

Table 1: Accuracy and memory consumption of LOZO and MeZO with their respective variants.

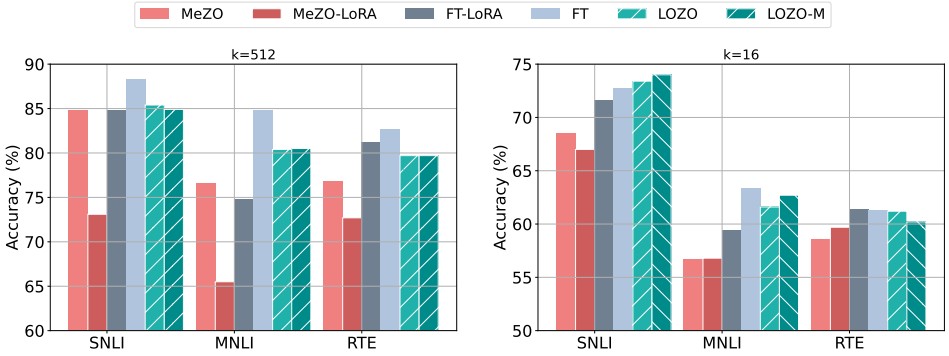

Figure 2: Comparison of the performance of different algorithms on RoBERTa-large across three tasks (SNLI, MNLI, and RTE), with the left panel corresponding to $k = 512$ and the right panel corresponding to $k = 16$. Detailed numerical results are provided in Table 8.

| Task | SST-2 | RTE | CB | BoolQ | WSC | WiC | MultiRC | COPA | ReCoRD | SQuAD | DROP |
|---|---|---|---|---|---|---|---|---|---|---|---|
| Zero-shot | 58.8 | 59.6 | 46.4 | 59.0 | 38.5 | 55.0 | 46.9 | 80.0 | 81.0 | 46.2 | 14.6 |
| ICL | 87.0 | 62.1 | 57.1 | 66.9 | 39.4 | 50.5 | 53.1 | 87.0 | **82.3** | 75.9 | 29.5 |
| MeZO | 91.3 | 68.2 | 66.1 | 68.1 | 61.5 | 59.4 | 59.4 | 88.0 | 81.3 | 81.8 | **31.3** |
| MeZO-LoRA | 89.6 | 67.9 | 67.8 | **73.5** | **63.5** | 60.2 | 61.3 | 84.0 | 81.5 | 82.1 | **31.3** |
| LOZO | **91.7** | **70.4** | **69.6** | 71.9 | **63.5** | **60.8** | **63.0** | **89.0** | 81.3 | **84.9** | 30.7 |
| FT | 91.8 | 70.9 | 84.1 | 76.9 | 63.5 | 70.1 | 71.1 | 79.0 | 74.1 | 84.9 | 31.3 |

Table 2: Experiments on OPT-13B (with 1000 examples). ICL: in-context learning; FT: full fine-tuning with Adam. The best results are shown in **bold** except for FT.

## 5.2 LARGE AUTOREGRESSIVE LANGUAGE MODELS

To further evaluate the effectiveness of LOZO on large language models, we extend our study to the OPT models (Zhang et al., 2022a) with billions of parameters (13B, 30B and 66B). The results are presented in Table 2 and Table 3.

**Compared to MeZO and MeZO-LoRA, LOZO demonstrates a clear improvement across the majority of datasets.** For instance, LOZO achieves notable performance gains on OPT-30B and OPT-66B, as presented in Table 3, outperforming MeZO and other baselines in most tasks. Furthermore, the results in Table 2 indicate that LOZO not only surpasses MeZO and MeZO-LoRA but also approaches the performance exhibited by FT on most cases.

**LOZO yields faster convergence rates across different model scales, including 13B, 30B and 66B.** As illustrated in Figure 3, the proposed method consistently achieves faster convergence across various datasets and model scales. For example, in the WIC dataset with the OPT-66B configuration, the LOZO algorithm requires only half the number of training epochs to achieve the same training loss as that of the MeZO method, while simultaneously exhibiting smaller loss oscillations.

| Task | SST-2 | RTE | BoolQ | WSC | WiC | SQuAD |
|---|---|---|---|---|---|---|
| 30B zero-shot | 56.7 | 52.0 | 39.1 | 38.5 | 50.2 | 46.5 |
| 30B ICL | 81.9 | **66.8** | 66.2 | 56.7 | 51.3 | 78.0 |
| 30B MeZO | 90.7 | 64.3 | 68.2 | 63.5 | 56.3 | **86.1** |
| 30B LOZO | **92.8** | 65.3 | **72.3** | **64.4** | **57.2** | 85.6 |
| 66B zero-shot | 57.5 | 67.2 | 66.8 | 43.3 | 50.6 | 48.1 |
| 66B ICL | 89.3 | 65.3 | 62.8 | 52.9 | 54.9 | 81.3 |
| 66B MeZO | 92.0 | 71.5 | 73.8 | **64.4** | 57.8 | 84.0 |
| 66B LOZO | **92.5** | **74.0** | **74.5** | 63.5 | **59.4** | **85.8** |

Table 3: Experiments on OPT-30B and OPT-66B on SuperGLUE benchmark. Our results show that LOZO is superior on most tasks compared to the other baselines. The best results are shown in **bold**.

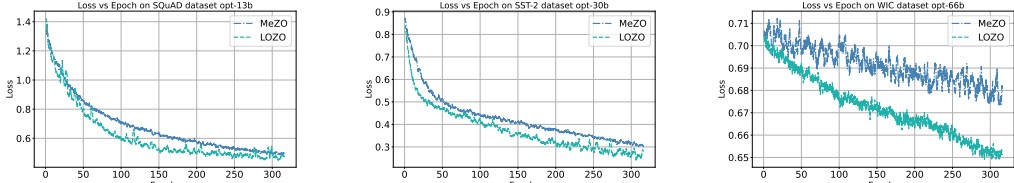

Figure 3: *Left:* Loss curves of OPT-13B on SQuAD dataset. *Middle:* Loss curves of OPT-30B on SST-2 dataset. *Right:* Loss curves of OPT-66B on WIC dataset.

## 6 CONCLUSION AND LIMITATIONS

This paper introduces the LOZO and LOZO-M algorithms, which are novel zeroth-order (ZO) methods for fine-tuning language models. Specifically, the LOZO algorithm employs a gradient estimator with a low-rank structure, closely mirroring the true gradient in first-order (FO) methods. We further demonstrate that the LOZO algorithm is equivalent to a ZO subspace method, forming the basis for our convergence results. By combining LOZO with the commonly used momentum technique, we develop the LOZO-M algorithm, which incurs almost no additional memory overhead. Both LOZO and LOZO-M achieve improved performance compared to the vanilla ZO-SGD method.

One limitation of our work is the challenge of designing a method that integrates LOZO with the Adam optimizer without incurring additional memory costs. Additionally, minor fluctuations in the loss are observed towards the end of the training process, potentially due to the lazy sampling strategy. Addressing these issues is left for future work.

ACKNOWLEDGMENTS

The computational resource was provided by the Center for Intelligent Computing, Great Bay Institute for Advanced Study, Dongguan, China. This work was supported in part by National Key Research and Development Program of China under the grant numbers 2024YFA1012902 and 2024YFA1012903, and the National Natural Science Foundation of China under the grant numbers 12331010, 12288101, 12401408 and W2441021. We also thank the anonymous reviewers for their valuable feedback.

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

## A  MORE COMMENTS

### A.1  UNBIASENESS OF LGE

In the proposed LOZO algorithm, we employ LGE to approximate the gradient. The following proposition demonstrates that the LGE scheme is unbiased as $\epsilon \to 0$.

**Proposition A.1.** *If $F(\boldsymbol{X};\xi)$ is differentiable with respect to $\boldsymbol{X}$, the LGE in (5) is an unbiased estimator of $\nabla_{X_\ell} F(\boldsymbol{X};\xi)$ when $\epsilon \to 0$.*

*Proof.* For simplicity, we omit the random variable $\xi$ in this proof. Under the differentiability assumption, for any set of real matrices $\{\Delta X_\ell\}_{\ell=1}^{\mathcal{L}}$ with the same dimensions as $\boldsymbol{X}$, we have

$$\lim_{\epsilon \to 0} \frac{F(\{X_\ell + \epsilon\Delta X_\ell\}_{\ell=1}^{\mathcal{L}}) - F(\boldsymbol{X}) - \sum_{\ell=1}^{\mathcal{L}}\langle\nabla_{X_\ell}F(\boldsymbol{X}), \epsilon\Delta X_\ell\rangle}{\epsilon} = 0.$$

Now, since $\hat{\nabla}_{X_\ell}F(\boldsymbol{X})$ from (5) can be written as

$$\left[\begin{array}{l} \dfrac{F(\{X_k + \epsilon U_k V_k^T\}_{k=1}^{\mathcal{L}}) - F(\boldsymbol{X}) - \sum_{k=1}^{\mathcal{L}}\langle\nabla_{X_k}F(\boldsymbol{X}), \epsilon U_k V_k^T\rangle}{2\epsilon} \\[2mm] -\dfrac{F(\{X_k - \epsilon U_k V_k^T\}_{k=1}^{\mathcal{L}}) - F(\boldsymbol{X}) - \sum_{k=1}^{\mathcal{L}}\langle\nabla_{X_k}F(\boldsymbol{X}), -\epsilon U_k V_k^T\rangle}{2\epsilon} \\[2mm] +\dfrac{1}{\epsilon}\sum_{k=1}^{\mathcal{L}}\langle\nabla_{X_k}F(\boldsymbol{X}), \epsilon U_k V_k^T\rangle \end{array}\right]\frac{U_\ell V_\ell^T}{r_\ell},$$

we have

$$\lim_{\epsilon \to 0}\mathbb{E}\left[\hat{\nabla}_{X_\ell}F(\boldsymbol{X})\right] = \frac{1}{r_\ell}\mathbb{E}\left[\sum_{k=1}^{\mathcal{L}}\langle\nabla_{X_k}F(\boldsymbol{X}), U_k V_k^T\rangle U_\ell V_\ell^T\right].$$

Since all elements of $\{U_\ell, V_\ell\}_{\ell=1}^{\mathcal{L}}$ are i.i.d. Gaussian variables, we can further deduce that

$$\lim_{\epsilon \to 0}\mathbb{E}\left[\hat{\nabla}_{X_\ell}F(\boldsymbol{X})\right] = \frac{1}{r_\ell}\mathbb{E}\left[\langle\nabla_{X_\ell}F(\boldsymbol{X}), U_\ell V_\ell^T\rangle U_\ell V_\ell^T\right].$$

The element at row $i$ and column $j$ of this expression is

$$\begin{aligned} &\frac{1}{r_\ell}\mathbb{E}\left[\langle\nabla_{X_\ell}F(\boldsymbol{X}), U_\ell V_\ell^T\rangle U_\ell V_\ell^T\right]_{ij} \\ &= \frac{1}{r_\ell}\mathbb{E}\left[\sum_{p=1}^{m_\ell}\sum_{q=1}^{n_\ell}\frac{\partial F(\boldsymbol{X})}{\partial[X_\ell]_{pq}}\sum_{k=1}^{r_\ell}[U_\ell]_{pk}[V_\ell]_{qk} \cdot \sum_{s=1}^{r_\ell}[U_\ell]_{is}[V_\ell]_{js}\right] \\ &= \frac{1}{r_\ell}\frac{\partial F(\boldsymbol{X})}{\partial[X_\ell]_{ij}}\mathbb{E}\left[\sum_{k=1}^{r_\ell}[U_\ell]_{ik}^2[V_\ell]_{jk}^2\right] = \frac{\partial F(\boldsymbol{X})}{\partial[X_\ell]_{ij}}. \end{aligned} \tag{21}$$

Therefore, we conclude that

$$\lim_{\epsilon \to 0}\mathbb{E}\left[\hat{\nabla}_{X_\ell}F(\boldsymbol{X})\right] = \nabla_{X_\ell}F(\boldsymbol{X}).$$

$\square$

### A.2 EQUIVALENCE BEWTWEEN LOZO AND ZO SUBSPACE METHOD

We present the detailed proof of the equivalence between LOZO algorithm (9) and the ZO subspace method (14). Let $\boldsymbol{X}^t$ be the $t$-th iteration point of LOZO, and let $\tilde{\boldsymbol{X}}^{(k)}$ be the $k$-th iteration point of the outer loop in the ZO subspace method (14b). We now show that $\boldsymbol{X}^{k\nu} = \tilde{\boldsymbol{X}}^{(k)}$ holds if the initialization of both algorithms is the same, i.e., $\boldsymbol{X}^0 = \tilde{\boldsymbol{X}}^{(0)}$.

We now introduce $\boldsymbol{Y}^{(k,s)} := \tilde{\boldsymbol{X}}^{(k)} + \boldsymbol{B}^{(k,s)}(\boldsymbol{V}^{(k)})^T$. When the step size satisfies $\boldsymbol{\gamma} = \alpha/\boldsymbol{r}$, using the update rules for $\boldsymbol{B}$ from (14a) and (15), we can derive the update rule for $\boldsymbol{Y}$, which is:

$$\boldsymbol{Y}^{(k,s+1)} = \boldsymbol{Y}^{(k,s)} - \boldsymbol{\gamma}\frac{F(\boldsymbol{Y}^{(k,s)} + \epsilon\boldsymbol{U}^{(k,s)}(\boldsymbol{V}^{(k)})^T) - F(\boldsymbol{Y}^{(k,s)} - \epsilon\boldsymbol{U}^{(k,s)}(\boldsymbol{V}^{(k)})^T)}{2\epsilon}\boldsymbol{U}^{(k,s)}(\boldsymbol{V}^{(k)})^T$$

$$= \boldsymbol{Y}^{(k,s)} - \alpha \cdot \mathrm{LGE}(\boldsymbol{Y}^{(k,s)}, \boldsymbol{U}^{(k,s)}, \boldsymbol{V}^{(k)}, \boldsymbol{r}, \epsilon, \xi^{(k,s)}), \quad \forall s \in \{0, \cdots, \nu-1\}. \quad (22)$$

This update rule for $\boldsymbol{Y}$ aligns with that for $\boldsymbol{X}$ in the LOZO algorithm. Thus, once $\boldsymbol{Y}^{(k,0)} = \tilde{\boldsymbol{X}}^{(k)} = \boldsymbol{X}^{k\nu}$, it follows that $\boldsymbol{Y}^{(k,s)} = \tilde{\boldsymbol{X}}^{(k)} + \boldsymbol{B}^{(k,s)}(\boldsymbol{V}^{(k)})^T = \boldsymbol{X}^{k\nu+s}$ for all $s \in \{0, 1, \ldots, \nu\}$. Therefore, $\boldsymbol{Y}^{(k,\nu)} = \tilde{\boldsymbol{X}}^{(k)} + \boldsymbol{B}^{(k,\nu)}(\boldsymbol{V}^{(k)})^T = \tilde{\boldsymbol{X}}^{(k+1)} = \boldsymbol{X}^{(k+1)\nu}$. Thus, the relation $\tilde{\boldsymbol{X}}^{(k)} = \boldsymbol{X}^{k\nu}$ holds for any $k$, given that $\tilde{\boldsymbol{X}}^{(0)} = \boldsymbol{X}^0$.

### A.3 DETAILS OF LOZO-M

We present the detailed LOZO-M algorithm in Algorithm 2. Compared to the LOZO algorithm, the main difference is the addition of the momentum term update.

---

**Algorithm 2:** Low-rank ZO-SGD with Momentum (LOZO-M)

---

**Input:** parameters $\boldsymbol{X}$, loss function $F(\boldsymbol{X};\xi)$, step budget $T$, perturbation scale $\epsilon$, learning rate $\alpha$, momentum parameter $\beta$, sample interval $\nu$ and rank $\{r_\ell\}$.

**foreach** $X_\ell \in \boldsymbol{X}$ **do**
$\quad$ $N_\ell \leftarrow 0$ ; $\qquad\qquad\qquad\qquad\qquad\qquad$ // Initialize momentum

**for** $t = 0, \ldots, T-1$ **do**
$\quad$ **foreach** $X_\ell \in \boldsymbol{X}$ **do**
$\quad\quad$ Sample $U_\ell \in \mathbb{R}^{m_\ell \times r_\ell}$ from the standard normal distribution;
$\quad\quad$ **if** $t \bmod \nu = 0$ **then**
$\quad\quad\quad$ $M_\ell \leftarrow N_\ell V_\ell^T$;
$\quad\quad\quad$ Sample $V_\ell \in \mathbb{R}^{n_\ell \times r_\ell}$ from the standard normal distribution; ; $\quad$ // Resample $V_\ell$
$\quad\quad\quad$ $N_\ell \leftarrow \frac{1}{n_\ell}M_\ell V_\ell$; ; $\quad$ // Project momentum onto the new subspace

$\quad$ $\boldsymbol{X} \leftarrow \mathrm{Perturbation}(\boldsymbol{X}, \epsilon, \{U_\ell, V_\ell\})$;
$\quad$ $F_+ \leftarrow F(\boldsymbol{X};\xi)$;
$\quad$ $\boldsymbol{X} \leftarrow \mathrm{Perturbation}(\boldsymbol{X}, -2\epsilon, \{U_\ell, V_\ell\})$;
$\quad$ $F_- \leftarrow F(\boldsymbol{X};\xi)$;
$\quad$ $\boldsymbol{X} \leftarrow \mathrm{Perturbation}(\boldsymbol{X}, \epsilon, \{U_\ell, V_\ell\})$; ; $\qquad\qquad\quad$ // Reset parameters
$\quad$ $c \leftarrow (F_+ - F_-)/2\epsilon$ ; $\qquad\qquad\qquad$ // Calculate finite difference
$\quad$ **foreach** $X_\ell \in \boldsymbol{X}$ **do**
$\quad\quad$ $N_\ell \leftarrow \beta N_\ell + (1-\beta) \cdot cU_\ell$; ; $\qquad$ // Update momentum in place
$\quad\quad$ $X_\ell \leftarrow X_\ell - \alpha(N_\ell V_\ell^T/r_\ell)$; ; $\qquad$ // Update parameters in place

**Function** Perturbation$(\boldsymbol{X}, \epsilon, \{U_\ell, V_\ell\})$**:**
$\quad$ **foreach** $X_\ell \in \boldsymbol{X}$ **do**
$\quad\quad$ $X_\ell \leftarrow X_\ell + \epsilon U_\ell V_\ell^T$; ; $\qquad\qquad$ // Modify parameters in place
$\quad$ **return** $\boldsymbol{X}$;

---

## B CONVERGENCE ANALYSIS

In this section, we present the convergence analysis of the LOZO algorithm and provide a detailed proof of Theorem 4.4. Without loss of generality, we focus on the case where the number of layers is

$\mathcal{L} = 1$. Consequently, the problem (1) reduces to $\min_X f(X) := \mathbb{E}_\xi[F(X; \xi)]$, where $X \in \mathbb{R}^{m \times n}$. To simplify the notation, we define the following terms:

$$G_{X,V}(B; \xi) := F(X + BV^T; \xi), \quad g_{X,V}(B) := f(X + BV^T),$$

$$\hat{\nabla} G_{X,V}(B; \xi) := \frac{G_{X,V}(B + \epsilon U) - G_{X,V}(B - \epsilon U)}{2\epsilon} U.$$

With these definitions, the subspace minimization problem (12) can be reformulated as follows:

$$\min_B g_{X,V}(B) = \mathbb{E}_\xi[G_{X,V}(B; \xi)]. \tag{23}$$

As demonstrated in Section 4.2, our proposed LOZO algorithm is equivalent to solving the subproblem (23) using ZO-SGD with the LGE scheme for $\nu$ steps, followed by an update of the weight matrix. Specifically, by applying the following update rule:

$$B^{(k,s+1)} = B^{(k,s)} - \frac{\alpha}{r} \hat{\nabla} G_{\tilde{X}^{(k)}, V^{(k)}}(B^{(k,s)}; \xi^{(k,s)}), \quad \forall s \in \{0, 1, \dots, \nu - 1\}, \tag{24a}$$

$$\tilde{X}^{(k+1)} = \tilde{X}^{(k)} + B^{(k,\nu)}(V^{(k)})^T, \tag{24b}$$

it follows that $X^{k\nu} = \tilde{X}^{(k)}$ for any $k$, where $X^{k\nu}$ represents the $k\nu$-th iteration point of the LOZO algorithm. For the remainder of the proof, we will use $X^{k\nu}$ in place of $\tilde{X}^{(k)}$ in (24).

Under Assumption 4.2, the following properties hold, which can be straightforwardly derived:

$$\|AV^T\|_F = \sqrt{\text{tr}(AV^TVA^T)} = \sqrt{n}\|A\|_F,$$

$$\|V\|_2 = \sqrt{\|V^TV\|_2} = \sqrt{n},$$

where $A \in \mathbb{R}^{m \times n}$ is any matrix. For simplicity, we will not explicitly reference these properties when they are used.

To construct the convergence result, our analysis is divided into two parts. First, we analyze the convergence of ZO-SGD (24a) for solving (23) with fixed $X$ and $V$. Next, we assess the impact of updating $X$ and resampling $V$, and establish the global convergence result for LOZO algorithm.

To begin the first part, we introduce some preliminary lemmas. All of these lemmas assume fixed $X$ and $V$, so we will omit the subscripts of $g_{X,V}(B)$ and $G_{X,V}(B; \xi)$ when there is no risk of confusion. The following lemma establishes the desirable properties of the objective function in (23), which are necessary for the convergence of the iteration given in (24a).

**Lemma B.1.** *Under Assumptions 4.1 and 4.2, the following properties hold:*

- *The function $G_{X,V}(B; \xi)$ is uniformly $\tilde{L}$-smooth with a constant $\tilde{L} = nL$.*

- *$\nabla G_{X,V}(B; \xi)$ is an unbiased estimator of $\nabla g_{X,V}(B)$, and its variance is bounded by*

$$\mathbb{E}\|\nabla G_{X,V}(B; \xi) - \nabla g_{X,V}(B)\|_F^2 \le \tilde{\sigma}^2,$$

*where $\tilde{\sigma}^2 = n\sigma^2$.*

*Proof.* Given any $\xi$, since $F(X, \xi)$ is differentiable and $X + UV^T$ is a linear function of $U$, the function $G_{X,V}(U; \xi)$ is also differentiable, and we have:

$$\nabla G_{X,V}(B, \xi) = \nabla F(X + BV^T)V.$$

Thus, it follows that:

$$\begin{aligned}
\|\nabla G_{X,V}(B_1, \xi) - \nabla G_{X,V}(B_2, \xi)\|_F &= \|\nabla F(X + B_1 V^T, \xi)V - \nabla F(X + B_2 V^T, \xi)V\|_F \\
&\le \|\nabla F(X + B_1 V^T, \xi) - \nabla F(X + B_2 V^T, \xi)\|_F \|V\|_2 \\
&\le L\|(B_1 - B_2)V^T\|_F \|V\|_2 \\
&\le nL\|B_1 - B_2\|_F.
\end{aligned}$$

The second property holds because

$$
\begin{aligned}
&\mathbb{E}_\xi \|\nabla G_{X,V}(U;\xi) - \nabla g_{X,V}(U)\|_F^2 \\
&= \mathbb{E}_\xi \big\| \big[\nabla F(X + UV^T;\xi) - \nabla f(X + UV^T)\big]V \big\|_F^2 \\
&\leq \mathbb{E}_\xi \big\|\nabla F(X + UV^T;\xi) - \nabla f(X + UV^T)\big\|_F^2 \|V\|_2^2 \\
&\leq \sigma^2 \|V\|_2^2 = n\sigma^2.
\end{aligned}
$$

$\square$

The following lemma, which bounds the second moment of the gradient estimator, is necessary for proving the convergence of ZO-SGD.

**Lemma B.2.** *For the gradient estimator $\hat{\nabla} G_{X,V}(B;\xi)$, the following bound holds:*

$$
\mathbb{E}_U \|\hat{\nabla} G_{X,V}(B;\xi)\|_F^2 \leq 6mr\|\nabla G_{X,V}(B;\xi)\|_F^2 + 64\tilde{L}^2 m^3 r^3 \epsilon^2.
$$

*Proof.* The triangle inequality provides the following bound:

$$
\mathbb{E}_U \|\hat{\nabla} G(B;\xi)\|_F^2 \leq 2\mathbb{E}_U \|\hat{\nabla} G(B;\xi) - \langle \nabla G(B;\xi), U\rangle U\|_F^2 + 2\mathbb{E}_U \|\langle \nabla G(B;\xi), U\rangle U\|_F^2.
$$

To bound the first term, we use Assumption 4.1, leading to the following inequalities:

$$
|G(B + \epsilon U;\xi) - G(B;\xi) - \epsilon\langle \nabla G(B;\xi), U\rangle| \leq \frac{\tilde{L}\epsilon^2}{2}\|U\|_F^2,
$$

$$
|G(B - \epsilon U;\xi) - G(B;\xi) + \epsilon\langle \nabla G(B;\xi), U\rangle| \leq \frac{\tilde{L}\epsilon^2}{2}\|U\|_F^2.
$$

Combining these two inequalities, we obtain:

$$
\left| \frac{G(B + \epsilon U;\xi) - G(B - \epsilon U;\xi)}{2\epsilon} - \langle \nabla G(B;\xi), U\rangle \right| \leq \frac{\tilde{L}\epsilon}{2}\|U\|_F^2.
$$

Multiplying by $U$ on both sides and taking the expectation with respect to $U$ yields:

$$
\mathbb{E}_U \|\hat{\nabla} G(B;\xi) - \langle \nabla G(B;\xi), U\rangle U\|_F^2 \leq \frac{\tilde{L}^2\epsilon^2}{4}\mathbb{E}_U \|U\|_F^6 \leq \frac{\tilde{L}^2(mr+4)^3\epsilon^2}{4} \leq 32\tilde{L}^2 m^3 r^3 \epsilon^2.
$$

In the final inequality, we use the fact $\mathbb{E}_U \|U\|_F^6 = mr(mr+2)(mr+4)$. The second term can be calculated directly, giving us:

$$
\mathbb{E}_U \|\langle \nabla G(B;\xi), U\rangle U\|_F^2 = (mr+2)\|\nabla G(B;\xi)\|_F^2 \leq 3mr\|\nabla G(B;\xi)\|_F^2.
$$

Combining these two inequalities completes the proof. $\square$

We now introduce the Gaussian smoothing function as follows:

$$
g_{X,V}^\epsilon(B) := \mathbb{E}_U[g_{X,V}(B + \epsilon U)] = \frac{1}{\kappa}\int g_{X,V}(B + \epsilon U)e^{-\frac{1}{2}\|U\|_F^2}dU.
$$

The following lemma outlines several properties of the Gaussian smoothing function.

**Lemma B.3** (Section 2 in (Nesterov & Spokoiny, 2017)). *For the Gaussian smoothing function $g_{X,V}^\epsilon(B)$, the following properties hold:*

- $\mathbb{E}_{U,\xi}[\hat{\nabla} G_{X,V}(B;\xi)] = \nabla g_{X,V}^\epsilon(B).$

- $g_{X,V}^\epsilon(B)$ *is $\tilde{L}$-smooth.*

- $|g_{X,V}^\epsilon(B) - g_{X,V}(B)| \leq \frac{\tilde{L}mr\epsilon^2}{2}.$

- $\|\nabla g_{X,V}^\epsilon(B) - \nabla g_{X,V}(B)\|_F^2 \leq \tilde{L}^2 mr\epsilon^2.$

*Proof.* We only prove the last claim. The remaining claims and their proofs can be found in (Nesterov & Spokoiny, 2017).

$$\|\nabla g^\epsilon(B) - \nabla g(B)\|_F^2 = \|\mathbb{E}_U(\nabla g(B + \epsilon U) - \nabla g(B))\|_F^2$$
$$\leq \mathbb{E}_U \|\nabla g(B + \epsilon U) - \nabla g(B)\|_F^2$$
$$\leq \tilde{L}^2 \epsilon^2 \mathbb{E}_U \|U\|_F^2 = \tilde{L}^2 m r \epsilon^2.$$

In the first inequality, we apply Jensen's inequality, and the second inequality derives from the $\tilde{L}$-smoothness of $g(B)$. $\qquad\square$

Now we are able to establish the convergence for solving problem (23) using the update rule (24a). The following lemma bounds the expected difference between any $B^t$ and $B^0$.

**Lemma B.4.** *If the step size condition $\frac{16\tilde{L}^2 m \nu \alpha^2}{r} \leq \frac{1}{\nu - 1}$ is satisfied, then for any $t \leq \nu$, the following inequality holds:*

$$\mathbb{E}_{U,\xi} \|B^t - B^0\|_F^2 \leq \frac{64 m \nu^2 \alpha^2}{r} \mathbb{E}_{U,\xi} \|\nabla g(B^0)\|_F^2 + \frac{24 m \nu \alpha^2 \tilde{\sigma}^2}{r} + 264 \tilde{L}^2 m^3 r \epsilon^2 \nu^2 \alpha^2.$$

*Proof.* To simplify the notation, we use $\mathbb{E}$ to denote the expectation taken over both $U$ and $\xi$. By the triangle inequality, we have the following:

$$\mathbb{E}\|B^{t+1} - B^0\|_F^2 = \mathbb{E}\left\|B^t - \frac{\alpha}{r}\hat{\nabla}G(B^t; \xi^t) - B^0\right\|_F^2$$
$$= \mathbb{E}\left\|B^t - \frac{\alpha}{r}\nabla g^\epsilon(B^t) - B^0\right\|_F^2 + \frac{\alpha^2}{r^2}\mathbb{E}\|\hat{\nabla}G(B^t; \xi^t) - \nabla g^\epsilon(B^t)\|_F^2$$
$$\leq \left(1 + \frac{1}{\nu - 1}\right)\mathbb{E}\|B^t - B^0\|_F^2 + \frac{\nu\alpha^2}{r^2}\mathbb{E}\|\nabla g^\epsilon(B^\nu)\|_F^2 + \frac{\alpha^2}{r^2}\mathbb{E}\|\hat{\nabla}G(B^t; \xi^t)\|_F^2.$$

Next, we use Lemmas B.2 and B.3 to bound the last two terms:

$$\mathbb{E}\|B^{t+1} - B^0\|_F^2 \leq \left(1 + \frac{1}{\nu - 1}\right)\mathbb{E}\|B^t - B^0\|_F^2 + \frac{2\nu\alpha^2}{r^2}\mathbb{E}\|\nabla g(B^t)\|_F^2$$
$$+ \frac{6m\alpha^2}{r}\mathbb{E}\|\nabla G(B^t; \xi^t)\|_F^2 + 66\tilde{L}^2 m^3 r \epsilon^2 \nu\alpha^2$$
$$\leq \left(1 + \frac{1}{\nu - 1}\right)\mathbb{E}\|B^t - B^0\|_F^2 + \frac{8m\nu\alpha^2}{r}\mathbb{E}\|\nabla g(B^t)\|_F^2$$
$$+ \frac{6m\alpha^2\tilde{\sigma}^2}{r} + 66\tilde{L}^2 m^3 r \epsilon^2 \nu\alpha^2$$
$$\leq \left(1 + \frac{1}{\nu - 1} + \frac{16\tilde{L}^2 m \nu\alpha^2}{r}\right)\mathbb{E}\|B^t - B^0\|_F^2 + \frac{16m\nu\alpha^2}{r}\mathbb{E}\|\nabla g(B^0)\|_F^2$$
$$+ \frac{6m\alpha^2\tilde{\sigma}^2}{r} + 66\tilde{L}^2 m^3 r \epsilon^2 \nu\alpha^2.$$

Applying the step size condition $\frac{16\tilde{L}^2 m \nu\alpha^2}{r} \leq \frac{1}{\nu - 1}$, we obtain:

$$\mathbb{E}\|B^{t+1} - B^0\|_F^2 \leq \left(1 + \frac{2}{\nu - 1}\right)\mathbb{E}\|B^t - B^0\|_F^2 + \frac{16m\nu\alpha^2}{r}\mathbb{E}\|\nabla g(B^0)\|_F^2$$
$$+ \frac{6m\alpha^2\tilde{\sigma}^2}{r} + 66\tilde{L}^2 m^3 r \epsilon^2 \nu\alpha^2.$$

By induction, we have:

$$\mathbb{E}\|B^t - B^0\|_F^2 \leq \sum_{s=0}^{t-1}\left(1 + \frac{2}{\nu - 1}\right)^s \left(\frac{16m\nu\alpha^2}{r}\mathbb{E}\|\nabla g(B^0)\|_F^2 + \frac{6m\alpha^2\tilde{\sigma}^2}{r} + 66\tilde{L}^2 m^3 r \epsilon^2 \nu\alpha^2\right)$$
$$\leq \frac{64m\nu^2\alpha^2}{r}\mathbb{E}\|\nabla g(B^0)\|_F^2 + \frac{24m\nu\alpha^2\tilde{\sigma}^2}{r} + 264\tilde{L}^2 m^3 r \epsilon^2 \nu^2\alpha^2.$$

In the last inequality, we use the fact that $\sum_{s=0}^{t-1}\left(1 + \frac{2}{\nu - 1}\right)^s \leq 4\nu$ for $t \leq \nu$. $\qquad\square$

The following lemma provides a bound on the function value of (23) over $\nu$ iteration steps of the update rule (24a).

**Lemma B.5.** *If the step size satisfies the condition $32\tilde{L}m\nu\alpha \leq 1$, then the following holds:*

$$\mathbb{E}_{U,\xi}(g_{X,V}(B^\nu) - g_{X,V}(B^0)) \leq \left(-\frac{\nu\alpha}{4r} + \frac{18\tilde{L}m\nu^2\alpha^2}{r}\right)\mathbb{E}_{U,\xi}\|\nabla g_{X,V}(B^0)\|_F^2$$

$$+ \frac{7\tilde{L}m\tilde{\sigma}^2\nu\alpha^2}{r} + 2\tilde{L}mr\epsilon^2,$$

*where $\tilde{L}$ and $\tilde{\sigma}^2$ are defined in Lemma B.1.*

*Proof.* We continue to use $\mathbb{E}$ to denote the expectation taken over both $U$ and $\xi$. We begin with the following inequality, which is derived from the $\tilde{L}$-smoothness of $g^\epsilon(B)$:

$$\mathbb{E}g^\epsilon(B^\nu) - \mathbb{E}g^\epsilon(B^0) \leq \mathbb{E}\langle\nabla g^\epsilon(B^0), B^\nu - B^0\rangle + \frac{\tilde{L}}{2}\mathbb{E}\|B^\nu - B^0\|_F^2$$

$$\leq -\frac{\alpha}{r}\sum_{t=0}^{\nu-1}\mathbb{E}\langle\nabla g^\epsilon(B^0), \hat{\nabla}G(B^t;\xi^t)\rangle + \frac{\tilde{L}\alpha^2}{2r^2}\mathbb{E}\left\|\sum_{t=0}^{\nu-1}\hat{\nabla}G(B^t;\xi^t)\right\|_F^2.$$

For the first term on the right-hand side, we have:

$$-\frac{\alpha}{r}\sum_{t=0}^{\nu-1}\mathbb{E}\langle\nabla g^\epsilon(B^0), \hat{\nabla}G(B^t;\xi^t)\rangle = -\frac{\alpha}{r}\sum_{t=0}^{\nu-1}\mathbb{E}\langle\nabla g^\epsilon(B^0), \nabla g^\epsilon(B^t)\rangle$$

$$= -\frac{\alpha}{r}\sum_{t=0}^{\nu-1}\mathbb{E}\langle\nabla g^\epsilon(B^0), \nabla g^\epsilon(B^t) - \nabla g^\epsilon(B^0) + \nabla g^\epsilon(B^0)\rangle$$

$$= -\frac{\nu\alpha}{r}\|\nabla g^\epsilon(B^0)\|_F^2 - \frac{\alpha}{r}\sum_{t=0}^{\nu-1}\mathbb{E}\langle\nabla g^\epsilon(B^0), \nabla g^\epsilon(B^t) - \nabla g^\epsilon(B^0)\rangle$$

$$\leq -\frac{\nu\alpha}{2r}\mathbb{E}\|\nabla g^\epsilon(B^0)\|_F^2 + \frac{\alpha}{2r}\sum_{t=0}^{\nu-1}\mathbb{E}\|\nabla g^\epsilon(B^t) - \nabla g^\epsilon(B^0)\|_F^2$$

$$\leq -\frac{\nu\alpha}{4r}\mathbb{E}\|\nabla g(B^0)\|_F^2 + \frac{\tilde{L}^2\alpha}{2r}\sum_{t=0}^{\nu-1}\mathbb{E}\|B^t - B^0\|_F^2 + \frac{\tilde{L}^2m\epsilon^2\nu\alpha}{2}.$$

The first inequality follows from $\langle a, b\rangle \leq \frac{\|a\|^2 + \|b\|^2}{2}$, and the final inequality holds due to Lemmas B.1 and B.3.

For the second term, we have:

$$\frac{\tilde{L}\alpha^2}{2r^2}\mathbb{E}\left\|\sum_{t=0}^{\nu-1}\hat{\nabla}G(B^t;\xi^t)\right\|_F^2 \leq \frac{\tilde{L}\alpha^2}{r^2}\mathbb{E}\left\|\sum_{t=0}^{\nu-1}\hat{\nabla}G(B^t;\xi^t) - \nabla g^\epsilon(B^t)\right\|_F^2 + \frac{\tilde{L}\nu\alpha^2}{r^2}\sum_{t=0}^{\nu-1}\mathbb{E}\|\nabla g^\epsilon(B^t)\|_F^2$$

$$= \frac{\tilde{L}\alpha^2}{r^2}\sum_{t=0}^{\nu-1}\mathbb{E}\left\|\hat{\nabla}G(B^t;\xi^t) - \nabla g^\epsilon(B^t)\right\|_F^2 + \frac{\tilde{L}\nu\alpha^2}{r^2}\sum_{t=0}^{\nu-1}\mathbb{E}\|\nabla g^\epsilon(B^t)\|_F^2$$

$$\leq \frac{\tilde{L}\alpha^2}{r^2}\sum_{t=0}^{\nu-1}\mathbb{E}\|\hat{\nabla}G(B^t,\xi^t)\|_F^2 + \frac{\tilde{L}\nu\alpha^2}{r^2}\sum_{t=0}^{\nu-1}\mathbb{E}\|\nabla g^\epsilon(B^t)\|_F^2$$

$$\leq \frac{6\tilde{L}m\alpha^2}{r}\sum_{t=0}^{\nu-1}\mathbb{E}\|\nabla G(B^t;\xi^t)\|_F^2 + 64\tilde{L}^3m^3r\epsilon^2\nu\alpha^2 + \frac{\tilde{L}\nu\alpha^2}{r^2}\sum_{t=0}^{\nu-1}\mathbb{E}\|\nabla g^\epsilon(B^t)\|_F^2$$

$$\leq \frac{8\tilde{L}m\nu\alpha^2}{r}\sum_{t=0}^{\nu-1}\mathbb{E}\|\nabla g(B^t)\|_F^2 + \frac{6\tilde{L}m\tilde{\sigma}^2\nu\alpha^2}{r} + 66\tilde{L}^3m^3r\epsilon^2\nu^2\alpha^2$$

$$\leq \frac{16\tilde{L}m\nu^2\alpha^2}{r}\mathbb{E}\|\nabla g(B^0)\|_F^2 + \frac{16\tilde{L}^3m\nu\alpha^2}{r}\sum_{t=0}^{\nu-1}\mathbb{E}\|B^t - B^0\|_F^2$$
$$+ \frac{6\tilde{L}m\tilde{\sigma}^2\nu\alpha^2}{r} + 66\tilde{L}^3m^3r\epsilon^2\nu^2\alpha^2.$$

The first equation holds due to the independence of $U^t$ and $\xi^t$ for each $t$, while the third and fourth inequalities follow from Lemmas B.2 and B.3, respectively. Combining the above results and considering the condition $32\tilde{L}m\nu\alpha \leq 1$, we obtain:

$$\mathbb{E}(g^\epsilon(B^\nu) - g^\epsilon(B^0)) \leq \left(-\frac{\nu\alpha}{4r} + \frac{16\tilde{L}m\nu^2\alpha^2}{r}\right)\mathbb{E}\|\nabla g(B^0)\|_F^2 + \frac{\tilde{L}^2\alpha}{r}\sum_{t=0}^{\nu-1}\mathbb{E}\|B^t - B^0\|_F^2$$
$$+ \frac{6\tilde{L}m\tilde{\sigma}^2\nu\alpha^2}{r} + 66\tilde{L}^3m^3r\epsilon^2\nu^2\alpha^2 + \frac{\tilde{L}^2m\epsilon^2\nu\alpha}{2}.$$

Applying Lemma B.4 and again considering the condition $32\tilde{L}m\nu\alpha \leq 1$, we have:

$$\mathbb{E}(g^\epsilon(B^\nu) - g^\epsilon(B^0)) \leq \left(-\frac{\nu\alpha}{4r} + \frac{18\tilde{L}m\nu^2\alpha^2}{r}\right)\mathbb{E}\|\nabla g(B^0)\|_F^2 + \frac{7\tilde{L}m\tilde{\sigma}^2\nu\alpha^2}{r} + \tilde{L}mr\epsilon^2.$$

Finally, by applying Lemma B.3 once again, we can complete our proof. □

Now we have established the bound for solving the subproblem (23). Next, we investigate the impact of updating $X$ and resampling $V$ and establish the convergence result for our proposed LOZO algorithm. This leads to the following theorem.

**Theorem B.6** (Theorem 4.4). *Under Assumptions 4.1 and 4.2, and assuming the step size $\alpha \leq \frac{1}{144Lmn\nu}$, when applying the proposed LOZO algorithm to solve problem (1), and letting $T = K\nu$, the following inequality holds:*

$$\frac{1}{K}\sum_{k=0}^{K-1}\mathbb{E}\|\nabla f(X^{k\nu})\|^2 \leq \frac{8\Delta_0}{T\alpha} + \frac{56Lmn^2\sigma^2\alpha}{r} + \frac{16Lmnr\epsilon^2}{\nu\alpha},$$

*where $\Delta_0 := f(X^0) - f^*$. Furthermore, if we choose*

$$\epsilon = \sqrt{\frac{\Delta_0\nu}{16TLmnr}}, \quad \alpha = \left(144Lmn\nu + \sqrt{\frac{56TLmn^2\sigma^2}{9\Delta_0 r}}\right)^{-1},$$

*then it holds that:*

$$\frac{1}{K}\sum_{k=0}^{K-1}\mathbb{E}\|\nabla f(X^{k\nu})\|^2 \leq 16\sqrt{\frac{1}{T}\left(\frac{7\Delta_0Lmn^2\sigma^2}{r}\right)} + \frac{2592\Delta_0Lmn\nu}{T}.$$

*Proof.* Recalling the update rule (24), it follows that

$$g_{X^{k\nu},V^{(k)}}(B^\nu) = f(X^{k\nu} + B^{(k,\nu)}(V^{(k)})^T) = f(X^{(k+1)\nu}), \quad g_{X^{k\nu},V^{(k)}}(B^0) = f(X^{k\nu}).$$

Moreover, note that $\nabla g_{X^{k\nu},V^{(k)}}(B^0) = \nabla f(X^{k\nu})V^{(k)}$. By applying Lemma B.5, we obtain:

$$\mathbb{E}_{U,\xi}(f(X^{(k+1)\nu}) - f(X^{k\nu})) \leq \left(-\frac{\nu\alpha}{4r} + \frac{18\tilde{L}m\nu^2\alpha^2}{r}\right)\mathbb{E}_{U,\xi}\|\nabla f(X^{k\nu})V^{(k)}\|_F^2$$
$$+ \frac{7\tilde{L}m\tilde{\sigma}^2\nu\alpha^2}{r} + 2\tilde{L}mr\epsilon^2.$$

Taking the expectation over $V^{(k)}$, and noting that $\mathbb{E}V^{(k)}(V^{(k)})^T = I$ (by Assumption 4.2), $V^{(k)}$ is independent of $X^{k\nu}$, we have:

$$\mathbb{E}(f(X^{(k+1)\nu}) - f(X^{k\nu})) \leq \left(-\frac{\nu\alpha}{4} + 18\tilde{L}m\nu^2\alpha^2\right)\mathbb{E}\|\nabla f(X^{k\nu})\|_F^2$$

$$+ \frac{7\tilde{L}m\tilde{\sigma}^2\nu\alpha^2}{r} + 2\tilde{L}mr\epsilon^2.$$

Note that $T = K\nu$. Rearranging the inequality above and summing over $K$ gives:

$$\left(\frac{1}{4} - 18\tilde{L}m\nu\alpha\right)\frac{1}{K}\sum_{k=1}^{K}\mathbb{E}\|\nabla f(X^{k\nu})\|_F^2 \leq \frac{\Delta_0}{T\alpha} + \frac{7\tilde{L}m\tilde{\sigma}^2\alpha}{r} + \frac{2\tilde{L}mr\epsilon^2}{\nu\alpha}.$$

Considering the step size condition $144\tilde{L}m\nu\alpha \leq 1$, and using $\tilde{L} = nL$ and $\tilde{\sigma}^2 = n\sigma^2$, we complete the proof. $\qquad\square$

## C EXPERIMENTAL DETAILS

### C.1 DATASETS

For RoBERTa-large, we evaluate the performance on six NLP tasks: SST-2 (Socher et al., 2013), SST-5 (Socher et al., 2013), SNLI (Bowman et al., 2015), MNLI (Williams et al., 2017), RTE (Dagan et al., 2005; Bar-Haim et al., 2006; Giampiccolo et al., 2007; Bentivogli et al., 2009), TREC (Voorhees & Tice, 2000). We adopt two settings: $k = 16$ and $k = 512$, which require 16 and 512 examples per class, respectively, during both the training and validation stages.

For OPT, we conduct experiments on the following datasets: SST-2, RTE, CB (De Marneffe et al., 2019), BoolQ (Clark et al., 2019), WSC (Levesque et al., 2012), WiC (Pilehvar & Camacho-Collados, 2018), MultiRC (Khashabi et al., 2018), COPA (Roemmele et al., 2011), ReCoRD (Zhang et al., 2018), SQuAD (Rajpurkar et al., 2016), DROP (Dua et al., 2019).

For the LLaMA model, we evaluate its performance on the SST-2, WiC, COPA, SQuAD, and Wino-Grande (Sakaguchi et al., 2021) datasets.

| Experiment | Hyperparameters | Values |
|---|---|---|
| LOZO | Batch size | 64 |
| | Learning rate (k=16) | 1e−6 |
| | Learning rate (k=512) | 2e−7 |
| | Rank ($r$) | $\{4, 8\}$ |
| | Interval ($\nu$) | $\{50, 100\}$ |
| | $\epsilon$ | 1e−3 |
| | Weight Decay | 0 |
| MeZO | Batch size | 64 |
| | Learning rate | $\{1e−7, 1e−6, 1e−5\}$ |
| | $\epsilon$ | 1e−3 |
| | Weight Decay | 0 |
| MeZO-LoRA | Batch size | 64 |
| | Learning rate | $\{1e−5, 5e−5, 1e−4\}$ |
| | $\epsilon$ | 1e−3 |
| | Weight Decay | 0.1 |
| | $(r, \alpha)$ | $(8, 16)$ |
| FT | Batch size | $\{8\}$ |
| | Learning rate | $\{1e−5, 3e−5, 5e−5\}$ |
| | Weight Decay | 0 |
| FT-LoRA | Batch size | $\{8\}$ |
| | Learning rate | $\{1e−4, 3e−4, 5e−4\}$ |
| | $(r, \alpha)$ | $(8, 16)$ |

Table 4: The hyperparameter grids used for RoBERTa-large experiments. The learning rate of the LOZO algorithm refers to $\alpha/r$. For LOZO-M, we introduce an additional parameter, $\beta_1$, which is searched over the range $\{0.5, 0.7, 0.9\}$.

| Experiment | Hyperparameters | Values |
|---|---|---|
| LOZO | Batch size | 16 |
|  | Learning rate | $\{1e{-}6, 1e{-}7\}$ |
|  | $\epsilon$ | $\{1e{-}3, 1e{-}4\}$ |
|  | Rank ($r$) | $\{1, 2, 4\}$ |
|  | Interval ($\nu$) | $\{50, 100\}$ |
| MeZO | Batch size | 16 |
|  | Learning rate | $\{1e{-}6, 1e{-}7\}$ or $\{1e{-}6, 5e{-}7, 1e{-}7\}$ for SQuAD and DROP |
|  | $\epsilon$ | $1e{-}3$ |
| MeZO-LoRA | Batch size | 16 |
|  | Learning rate | $\{1e{-}4, 5e{-}5\}$ or $\{1e{-}4, 5e{-}5, 1e{-}5\}$ for SQuAD and DROP |
|  | $\epsilon$ | $1e{-}2$ |
|  | $(r, \alpha)$ | $(8, 16)$ |
| FT | Batch size | 8 |
|  | Learning rate | $\{1e{-}5, 5e{-}5, 8e{-}5\}$ |

Table 5: The hyperparameter grids used for OPT experiments. The learning rate of the LOZO algorithm refers to $\alpha/r$.

| Experiment | Hyperparameters | Values |
|---|---|---|
| LOZO | Batch size | 16 |
|  | Learning rate (k=16) | $1e{-}7$ |
|  | Rank ($r$) | $\{2, 4\}$ |
|  | Interval ($\nu$) | $\{50, 100\}$ |
|  | $\epsilon$ | $1e{-}3$ |
|  | Weight Decay | 0 |
| MeZO | Batch size | 16 |
|  | Learning rate | $\{1e{-}7, 1e{-}6\}$ |
|  | $\epsilon$ | $1e{-}3$ |
|  | Weight Decay | 0 |
| FT | Batch size | $\{8\}$ |
|  | Learning rate | $\{1e{-}6, 1e{-}7\}$ |
| FT-LoRA | Batch size | $\{8\}$ |
|  | Learning rate | $\{1e{-}4, 1e{-}5\}$ |
|  | $(r, \alpha)$ | $(8, 16)$ |

Table 6: The hyperparameter grids used for LLaMA experiments. The learning rate of the LOZO algorithm refers to $\alpha/r$.

## C.2 HYPERPARAMETERS

In this section, we present the hyperparameter search grids to support the reproducibility of our experiments using the RoBERTa-large, OPT, and LLaMA models. Both MeZO and LOZO utilize a constant learning rate schedule, whereas FT and FT-LoRA adopt a linear learning rate schedule.

For MeZO, LOZO, and their respective variants, we conduct 100K training steps, evaluating the model every 10K steps for the RoBERTa-large model; 20K training steps with evaluations every 4K steps for the OPT model; and 20K training steps, evaluating every 500 steps for the LLaMA model. For all gradient-based algorithms, we adhere to the configurations described in (Malladi et al., 2023; Zhang et al., 2024).

## C.3 ABLATION STUDY

In this section, we explore how the choice of rank $r$ and the lazy update interval $\nu$ affect the performance of our algorithm. We begin by examining the impact of $\nu$ and $r$ using the SST-2, COPA and RTE datasets on the OPT-1.3b model. To illustrate the impact of different values of $r$ and $\nu$ on the

convergence rate, we present a plot of loss versus epochs in Figure 4. Also, we list the accuracy and training loss across different rank $r$ and $\nu$ for the three datasets in Table 7.

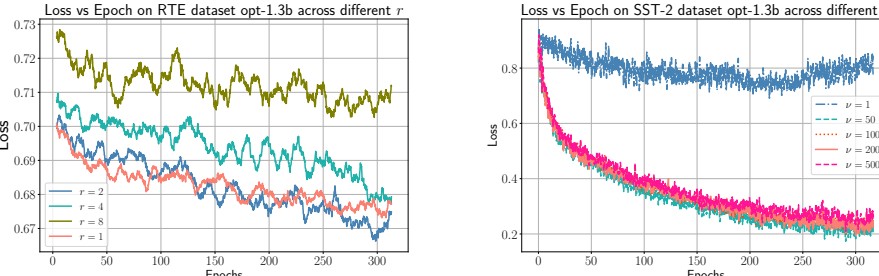

Figure 4: *Left:* Loss curves of OPT-1.3B on RTE dataset across different rank $r$. *Right:* Loss curves of OPT-1.3B on SST-2 dataset across different value $\nu$.

| $r$ | $\nu$ | SST-2 Accuracy | SST-2 loss | COPA Accuracy | COPA loss | RTE Accuracy | RTE loss |
|---|---|---|---|---|---|---|---|
| 1 | 50 | 88.1 | 0.45 | 73.0 | 1.93 | 56.7 | 0.68 |
|   | 100 | 89.0 | 0.46 | 74.0 | 2.18 | 56.7 | 0.68 |
| 2 | 1 | 55.0 | 0.79 | 74.0 | 2.58 | 50.9 | 0.70 |
|   | 50 | 93.0 | 0.37 | 74.0 | 2.04 | 61.0 | 0.69 |
|   | 100 | 92.1 | 0.37 | 71.0 | 2.05 | 58.1 | 0.68 |
|   | 200 | 92.7 | 0.37 | 77.0 | 2.05 | 62.1 | 0.67 |
|   | 500 | 91.7 | 0.37 | 75.0 | 2.05 | 62.8 | 0.67 |
| 4 | 50 | 91.3 | 0.35 | 76.0 | 1.99 | 57.4 | 0.69 |
|   | 100 | 92.0 | 0.35 | 75.0 | 1.97 | 57.8 | 0.69 |
| 8 | 50 | 88.5 | 0.48 | 71.0 | 2.03 | 55.0 | 0.71 |
|   | 100 | 88.9 | 0.45 | 73.0 | 2.03 | 56.3 | 0.71 |

Table 7: Performance and loss across different values of $r$ and $\nu$ on SST-2, COPA and RTE datasets.

**A small $\nu$ value can negatively impact convergence.** For datasets where the loss exhibits a significant decrease during fine-tuning, very small $\nu$ values can hinder the model's convergence, leading to degraded performance on the test datasets. For example, with a rank of 2 and $\nu = 1$, the final training loss reaches 0.79, nearly double that of other settings, as shown in Table 7. In addition, as shown in Figure 4, $\nu = 1$ exhibits different training dynamics compared to larger $\nu$ values, where the training loss either remains unchanged or even increases.

**A small $\nu$ value may not affect the model's performance on the test dataset.** In contrast, for datasets where the loss remains stable or decreases only slightly, the performance degradation caused by small $\nu$ values is minimal and less noticeable. The COPA dataset serves as a typical example, with the loss remaining nearly unchanged during training and unaffected by extremely small $\nu$ values. As shown in Table 7, the accuracy with a rank of 2 and $\nu = 1$ is comparable to that of other settings.

**A large rank $r$ can moderately slow down the training.** The left panel of Figure 4 demonstrates that a larger rank starts with a higher loss, requiring additional training epochs to reach the same loss compared to those small rank $r$.

**A small rank $r$ leads to a decline in model performance.** When setting the rank to $r = 1$, it consistently results in suboptimal performance across all three tasks.

# D MORE EXPERIMENTAL RESULTS

## D.1 ROBERTA-LARGE EXPERIMENTS

We present the complete results for RoBERTa-large. As shown in Table 8, our LOZO method and its variant, LOZO-M, outperform other gradient-free methods on almost all datasets.

| Task | SST-2 | SST-5 | SNLI | MNLI | RTE | TREC |
|------|-------|-------|------|------|-----|------|
| Type | sentiment | | natural language inference | | | topic |
| Zero-shot | 79.0 | 35.5 | 50.2 | 48.8 | 51.4 | 32.0 |
| Gradient-free methods: $k = 16$ | | | | | | |
| MeZO | 86.3 (8.0) | 40.8 (2.5) | 68.5 (4.2) | 56.7 (3.4) | 58.6 (9.5) | 62.4 (10.2) |
| MeZO-LoRA | **88.4 (1.6)** | 38.9 (2.0) | 67.0 (3.3) | 56.8 (1.0) | 59.7 (4.5) | 37.0 (5.1) |
| LOZO | 88.0 (5.8) | 41.1 (2.8) | 73.4 (4.0) | 61.6 (4.6) | **61.2 (9.1)** | 77.9 (7.4) |
| LOZO-M | 88.0 (5.8) | **42.9 (1.5)** | **74.0 (4.0)** | **62.7 (4.5)** | 60.2 (8.4) | **82.2 (5.2)** |
| Gradient-based methods: $k = 16$ | | | | | | |
| FT | 89.3 (5.3) | 44.0 (1.7) | 72.7 (5.7) | 63.4 (4.3) | 61.3 (5.3) | 83.7 (4.7) |
| FT-LoRA | 92.7 (0.9) | 45.5 (1.4) | 71.6 (4.5) | 59.4 (4.6) | 61.4 (6.9) | 75.8 (7.8) |
| Gradient-free methods: $k = 512$ | | | | | | |
| MeZO | 93.7 (0.4) | **53.9 (1.9)** | 84.8 (1.1) | 76.6 (0.8) | 76.8 (3.1) | 95.0 (0.4) |
| MeZO-LoRA | 91.7 (0.2) | 45.1 (1.4) | 73.1 (1.1) | 65.5 (0.9) | 72.7 (0.8) | 50.8 (1.9) |
| LOZO | 94.1 (0.7) | 53.0 (0.4) | **85.4 (0.8)** | 80.4 (1.0) | 79.7 (2.0) | **95.5 (0.4)** |
| LOZO-M | **94.3 (0.8)** | 52.6 (0.3) | 84.9 (1.1) | **80.5 (0.7)** | **79.7 (1.6)** | 95.5 (0.5) |
| Gradient-based methods: $k = 512$ | | | | | | |
| FT | 94.4 (0.6) | 55.7 (1.6) | 88.3 (0.8) | 84.8 (0.7) | 82.7 (1.1) | 97.2 (0.3) |
| FT-LoRA | 91.9 (2.1) | 52.4 (1.2) | 84.8 (0.6) | 74.8 (3.4) | 81.2 (1.6) | 96.1 (0.6) |

Table 8: Experimental results on RoBERTa-large (350M). All reported numbers are averaged accuracy (standard deviation). LOZO and LOZO-M outperforms MeZO and MeZO-LoRA by a considerable margin and approaches FT performance.

## D.2 OPT EXPERIMENTS

We have also applied the LOZO-M algorithm to the OPT-13B model, with results presented in Table 9. The numerical results indicate that incorporating the momentum technique further enhances the performance of LOZO across various tasks, even when applied to large model scales.

| Task | SST-2 | RTE | CB | WSC | COPA | SQuAD |
|------|-------|-----|-----|-----|------|-------|
| **LOZO** | 91.7 | 70.4 | **69.6** | 63.5 | 89.0 | **84.9** |
| **LOZO-M** | **92.5** | **73.6** | **69.6** | **64.4** | **90.0** | 83.3 |

Table 9: Comparison of the performance of LOZO and its momentum variant on OPT-13B.

| Task | RTE | | MultiRC | |
|------|-----|-----|---------|-----|
| | **Memory** | **Consumed GPUs** | **Memory** | **Consumed GPUs** |
| **LOZO** | 27.0 GB | 1× A800 | 26.9 GB | 1× A800 |
| **LOZO-M** | 27.4 GB | 1× A800 | 27.3 GB | 1× A800 |
| **MeZO** | 27.4 GB | 1× A800 | 27.3 GB | 1× A800 |
| **MeZO-M** | 51.7 GB | 1× A800 | 52.1 GB | 1× A800 |
| **FT-LoRA** | 79.0 GB | 1× A800 | 102.4 GB | 2× A800 |
| **FT** | 250.0 GB | 4× A800 | 315.2 GB | 4× A800 |

Table 10: Comparison of memory costs for LOZO, MeZO, their momentum variants, and gradient-based methods on OPT-13B.

In Table 10, we evaluate the minimum memory requirements for two datasets on OPT-13B, setting the per-device batch size to 1 to determine the minimum hardware requirements for running the model with different optimization algorithms. The results demonstrate that LOZO exhibits the lowest memory consumption, particularly when compared to FO methods. Notably, the momentum-enhanced LOZO variant (LOZO-M) incurs minimal additional memory overhead, unlike its counterpart, MeZO-M, which requires significantly more memory.

In Figure 5, we compare the convergence rates of our proposed LOZO algorithm with MeZO on two additional tasks. We also present a comparison of wall-clock times on GPUs. Despite having similar computational complexities, LOZO converges faster than MeZO, resulting in reduced wall-clock time.

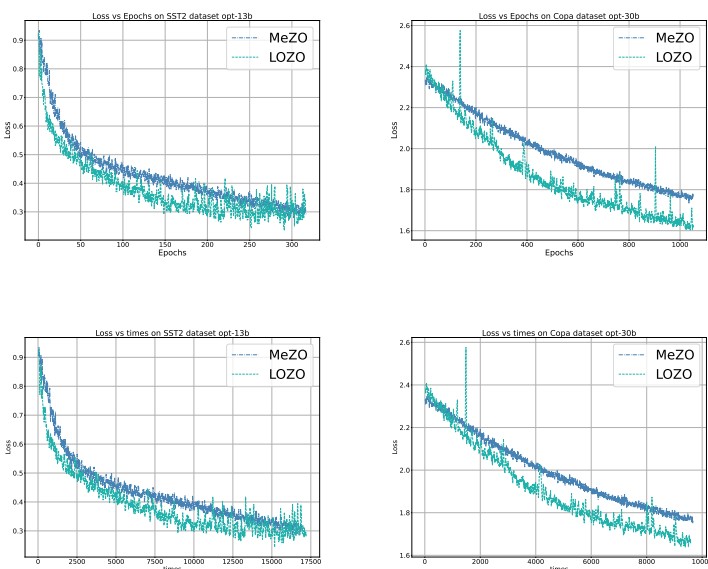

Figure 5: *Top:* Loss curves with respect to epochs for OPT-13B (SST2) and OPT-30B (Copa). *Bottom:* Loss curves with respect to time for the same configurations.

| Task | LLaMA-7B | | | | | LLaMA-13B | | LLaMA-70B |
|------|------|------|------|------|------|------|------|------|
| | SST-2 | WiC | COPA | SQuAD | WG | SST-2 | WG | WG |
| **LOZO** | **94.8** | **57.2** | 85.0 | **90.3** | **66.0** | **93.6** | **67.6** | **72.1** |
| **MeZO** | 91.6 | 56.3 | **86.0** | 90.0 | 64.3 | 92.1 | 67.2 | **72.1** |
| **FT-LoRA** | 95.1 | 69.4 | 84.0 | 91.2 | 70.9 | 95.5 | 76.6 | 50.4 |
| **FT** | 94.2 | 72.3 | 83.0 | 90.6 | 64.4 | 96.4 | 73.3 | - |

Table 11: Experimental results on LLaMA models of varying sizes. The superior results achieved by ZO methods are highlighted in **bold**. "WG" refers to the WinoGrande dataset. Due to limited computational resources, FT was not tested on LLaMA-70B.

| Task | LLaMA-7B | | LLaMA-70B | |
|------|----------|--|-----------|--|
| | **Memory** | **Consumed GPUs** | **Memory** | **Consumed GPUs** |
| LOZO | 14.1 GB | $1\times$ A800 | 135.5 GB | $2\times$ A800 |
| MeZO | 14.3 GB | $1\times$ A800 | 136.0 GB | $2\times$ A800 |
| FT-LoRA | 32.7 GB | $1\times$ A800 | 187.2 GB | $3\times$ A800 |
| FT | 281.6 GB | $4\times$ A800 | 640 + GB | $> 8\times$ A800 |

Table 12: Comparison of memory costs for LOZO, MeZO, and gradient-based methods on LLaMA models of varying scales for the MultiRC task with a per-device batch size of 1. Due to limited computational resources, the results for FT on LLaMA-70B are approximate.

### D.3 LLAMA EXPERIMENTS

We conducted experiments on the LLaMA model of varying sizes, comparing our proposed LOZO algorithm with MeZO and gradient-based algorithms. The results, presented in Table 11, demonstrate that our algorithm outperforms MeZO on most tasks.

In Table 12, we compare the memory requirements of LOZO, MeZO, and gradient-based methods with a per-device batch size of 1 for fine-tuning LLaMA-7B and LLaMA-70B models on the MultiRC dataset. The results demonstrate that as the model scale increases, the memory efficiency gap between ZO and gradient-based methods widens significantly.

