# OpenReview forum: "Enhancing Zeroth-order Fine-tuning for Language Models with Low-rank Structures"
_ICLR.cc/2025/Conference — ICLR 2025 Poster_

### Official Review · Reviewer_FzN7 · 2024-10-29

**Soundness:** 3
**Presentation:** 4
**Contribution:** 3
**Rating:** 8
**Confidence:** 4

**Summary:**

The paper studies the way to improve the performance of zeroth-order fine-tuning by doing perturbation in a low-rank space. The paper provide detailed theoretical proof for the convergence of the proposed algorithm and experimental results with similar setup as the previous ZO fine-tuning paper on model up to 30B.

**Strengths:**

The method proposed in this paper is interesting, which is inspired by the work on gradient low-rank structure. Also, the lazy sampling and LOZO-M methods are interesting to be considered. I quickly went through the proof for the convergence rate and it seems correct and reasonable, which provide solid support for the new subspace and lazy sampling method proposed in this paper.

**Weaknesses:**

Given the good theoretical foundation of this paper, my main concerns are about the parts of the experiment:
- I'm a bit confused about the total training steps for LoZO and other baselines like MeZO in the experiments. For example, in Figure 2, I'm not sure if the k represents epoch here as mentioned in the previous section, or similar to the MeZO paper, represents the number of shots. Also, I'm wondering what the total number of training steps here for LoZO and MeZO? Furthermore, it seems LOZO uses a different learning rate compared with MeZO, according to Table 4, which may make the comparison unfair.

- Still, for Fig. 2, I'm wondering why the MeZO-LoRA is performing worse than MeZO, as fewer trainable parameters should improve the ZO convergence rate according to the eq. (18) in the draft. I have extra two concerns here. First, this is different from the observation in Table 18 of MeZO paper, where MEZO-LoRA is performing better in most cases Second, I think it's reasonable that MEZO-LoRA fails to help on model larger than maybe 1B, where there are a lot of trainable parameters even with LoRA. But for small-size models like Roberta, MeZO-LoRA is reasonable to perform better with only 0.8 M parameters needing to be optimized. I would appreciate if the author could further explain this.

**Questions:**

My main questions are listed in the weakness section, here is a few of additional concerns:
- For Fig. 3, I have similar confusion about the total training steps. From my understanding, LOZO has an additional interval in each epoch. So LOZO has more training steps actually?
- I would appreciate it if the author could provide an evaluation loss vs. wall-time figure to demonstrate the effectiveness of the proposed method. Specifically, this would validate that the low-rank perturbation helps improve convergence speed with respect to training time, which is more critical than the number of training steps.
- The improvement in experiments is limited, considering the large variance of ZO method even between runs with different random seeds.
- How about LoZO-M perform on the large-scale model, just curious. I understand this is not the main purpose of your work and wondering if the momentum is becoming less important on large-scale models due to the large variance of ZO estimation.


Generally, I think this paper provides a solid method to improve the convergence of the ZO method. Further, clarifying experiments and including more discussion may benefit the reader of this paper.

---

> ### Author Response · Authors · 2024-11-22
>
> Thank you for your time and thoughtful feedback on our manuscript. Below are our detailed responses to the weaknesses and questions you raised:
>
> - Weaknesses:
>
> 1. In Figure 2, $k$ refers to the number of shots, consistent with the MeZO paper, rather than the $k$ used to denote the outer iteration number in Section 4.2. We apologize for the confusion caused by this reuse of notation.
>
>    Regarding the number of training steps, we measure them as the number of gradient estimations, consistent with the MeZO approach.
>
>    Regarding the learning rate, since MeZO and LOZO differ in their algorithmic structures, we tuned the optimal learning rate for each method separately. To address your concerns, we also conducted additional experiments to evaluate MeZO's performance with a learning rate of $2\times 10^{-7}$, which matches the learning rate used for LOZO on the RoBERTa-large model in the case \( k=512 \). The results are presented below.
>
>    | Task       | SST-2       | SST-5 |
>    |------------|--------------|----------------|
>    | **LOZO**   | 94.1         | 53.0           |
>    | **MeZO**   | 92.4         | 40.1           |
>
> 2. Thank you for this insightful question!
>
>    - Regarding the results in Table 18 of MeZO, unfortunately, we were unable to reproduce the exact results despite using the MeZO codebase and conducting a comprehensive grid search over hyperparameters. One possible explanation for the discrepancy is that, in the pre-trained RoBERTa-large model, a specific layer is randomly initialized rather than pre-trained, which may account for the differences. However, we would like to emphasize that, even when compared to the results reported in Table 18 of MeZO, our LOZO approach still achieves superior performance.
>
>    - We do not know whether your intuition that "for small-size models like Roberta, MeZO-LoRA is reasonable to perform better" is correct or not. Generally speaking, we believe fewer parameters may result in over-fitting and hence hurt generalization, which may explain why MeZO-LoRA performs worse than MeZO. In fact, according to Table 8, we find both LoRA and MeZO-LoRA perform poorly in the $k=512$ case but perform well in the $k=16$ case. We hypothesize that, with $k=512$, the larger data volume requires more trainable parameters to better handle the increased complexity.
>
>    Additionally, you pointed out that MeZO-LoRA has fewer trainable parameters, potentially leading to faster convergence as indicated by Equation (18) in the draft. However, Equation (18) pertains to the global convergence of $\min_X f(X)$, while LoRA only optimizes the adaptaors—namely, the low-rank adapters $A$ and $B$—and does not optimize the full parameter set $X$. In other words, MeZO-LoRA is not solving the same problem as MeZO (and LOZO), and hence the convergence rate in (18) cannot precisely reflect the convergence rate of MeZO-LoRA.

---

> > ### Author Response · Authors · 2024-11-22
> >
> > - Questions:
> >
> > 1. Although LOZO can be understood as a subspace optimization algorithm, it can also be implemented in a single-loop fashion, as described in Algorithm 1.
> > Comparing Algorithm 1 with MeZO, we observe that they share the same number of sampled data or gradient evaluations per iteration. As a result, it is fair to compare MeZO and LOZO by running them for the same number of training steps or epochs.
> >
> > 2. Thank you for the insightful suggestions. We have added two additional convergence speed tests for the LOZO algorithm on OPT-13B and 30B. These tests include plots of loss versus steps and loss versus wall-clock time on GPUs. Please refer to Figure 5 in Appendix D.2 of the revised manuscript.
> > Due to the similar computational complexity per iteration for LOZO and MeZO, LOZO requires less time to achieve the same loss level compared to MeZO.
> >
> > 3. We respectfully disagree that the improvment in experiments is limited. While it is true that LOZO performs similarly to MeZO on certain tasks, LOZO generally outperforms MeZO on most datasets. For instance, on the RTE dataset with RoBERTa-large and all sizes of OPT, LOZO consistently outperforms MeZO. Although zero-order optimization (ZO) algorithms can introduce variance in individual steps, this variance is mitigated over a large number of training steps.
> >
> >    To further support this claim, we repeat experiments on OPT-13B with the SST-2 and RTE datasets using three different seeds commonly adopted in the community. The results are shown below:
> >
> >    **SST-2:**
> >    |    | SEED 0   | SEED 42   | SEED 100  | Average  |
> >    |-------------|----------|-----------|-----------|----------|
> >    | **LOZO**    | 91.7     | 93.5      | 92.9      | 92.7     |
> >    | **MeZO**    | 91.3     | 91.1      | 91.5      | 91.3     |
> >
> >    **RTE:**
> >    |     | SEED 0   | SEED 42   | SEED 100  | Average  |
> >    |-------------|----------|-----------|-----------|----------|
> >    | **LOZO**    | 70.4     | 68.6      | 68.6      | 69.2     |
> >    | **MeZO**    | 68.2     | 65.3      | 65.3      | 66.5     |
> >
> >    As shown, LOZO consistently outperforms MeZO across all seeds, supporting our claim that LOZO's superior performance is not coincidental.
> >
> > 4. Thank you for your questions. We added comparisons between LOZO-M and vanilla LOZO on OPT-13B, with results presented in the table below:
> >
> >    | **Task**   | **SST-2** | **RTE** | **CB**  | **WSC** | **COPA** | **SQuAD** |
> >    |------------|-----------|---------|---------|---------|----------|-----------|
> >    | **LOZO**   | 91.7      | 70.4    | **69.6**| 63.5    | 89.0     | **84.9**  |
> >    | **LOZO-M** | **92.5**  | **73.6**| 69.6    | **64.4**| **90.0** | 83.3      |
> >
> >    From the table, we observe that LOZO-M achieves performance improvements on most tasks, though it does not always surpass vanilla LOZO.  These results are included in Table 9 of the revised manuscript.
> >
> > We hope these responses address your concerns. Please feel free to reach out with any further questions or feedback.

---

> ### Comment · Reviewer_FzN7 · 2024-11-25
> **Thanks for the response!**
>
> Thanks for the response from the authors, which solved most of my concerns. Generally, I think it's a good paper for improving the performance of zeroth-order optimization and I would like to increase my score to 8.
>
> However, I don't agree with some of the points in the response letter, even though these disagree will not influence the conclusion of this paper:
> > One possible explanation for the discrepancy is that, in the pre-trained RoBERTa-large model, a specific layer is randomly initialized rather than pre-trained, which may account for the differences.
>
> I think this paper follows the experiment setup in MEZO setup, which utilizes a prompted-based fine-tuning method. This means we are performing language modeling tasks during the fine-tuning process to solve the CLS problem, which use the language modeling head instead of the random initialized cls head.
>
> > In fact, according to Table 8, we find both LoRA and MeZO-LoRA perform poorly in the  K= 16 case but perform well in the
>  case. We hypothesize that, with k=512, the larger data volume requires more trainable parameters to better handle the increased complexity.
>
> We can see from the first-order result from the original LoRA paper, which uses a setup of a full training dataset, but the result between LoRA and Full model FT is still the same. This means, that even with a full training dataset, there is no obvious over-fitting. So there may be other reasons here that need to be explored.
>
> Again, I think these problems are not limited to the problem proposed in this paper, which will not influence the conclusion of this work. The method proposed achieves performance improvement with better time efficiency. Thus, I suggest the acceptance of this paper.

---

> > ### Author Response · Authors · 2024-12-02
> >
> > Thank you for taking the time to review our manuscript, providing valuable feedback once again, and offering positive comments on our work.
> >
> > **Regarding the first point**, we would like to clarify that upon downloading the pre-trained RoBERTa-large model from Hugging Face, we observed that the weight matrix of the pooler dense layer was initialized randomly, rather than being pre-trained. This random initialization may potentially affect the experimental results and could explain the discrepancies between our findings and those reported in the MeZO paper. However, to ensure a fair comparison, we have consistently used the same pooler dense layer weight matrix across all of our experiments.
> >
> > Regarding the CLS head and IM head, these layers are initialized during the fine-tuning process. Since we employed the same random seed for fine-tuning as used in the MeZO paper, we believe that the initialization of these layers does not contribute to any discrepancies in the final results.
> >
> > **For the second point**, in order to further investigate the performance issue of MeZO-LoRA when \(k=512\), we conducted additional experiments, increasing the number of iterations to four times that used in LOZO. The results demonstrate that as the number of iterations increases, the performance of MeZO-LoRA improves. Based on these findings, we now believe that the performance issue is not related to the limited number of trainable parameters, as you correctly suggested. Instead, it seems to be associated with the training process itself. We will continue to explore this issue in greater detail.
> >
> >
> >    |  Optimizer                   | SST-2    | SST-5     | SNLI      |
> >    |------------------------------|----------|-----------|-----------|
> >    | **LOZO**                    | 94.1      | 53.0      | 85.4      |
> >    | **MeZO-LoRA**               | 91.7     | 45.1      | 73.1      |
> >    | **MeZO-LoRA (4 x Iter)**    | 92.9      | 49.2      | 77.0      |
> >
> > We would be delighted to address any additional questions or concerns you may have.

---

### Official Review · Reviewer_4TZ5 · 2024-11-02

**Soundness:** 3
**Presentation:** 3
**Contribution:** 2
**Rating:** 6
**Confidence:** 4

**Summary:**

The manuscript presents a approach to parameter-efficient fine-tuning (PEFT) of large language models (LLMs) using zeroth-order (ZO) optimization. The authors propose a low-rank ZO gradient estimator and introduce a new algorithm called LOZO, which captures the low-rank gradient structure common in LLM fine-tuning. The paper provides convergence guarantees for LOZO, frames it as a subspace optimization method, and demonstrates its effectiveness through extensive experiments across various model sizes and downstream tasks.

**Strengths:**

Strengths:

1. theoretical guarantees: The manuscript introduces a low-rank ZO gradient estimator and algorithm (LOZO) that addresses the memory inefficiency issue associated with traditional first-order fine-tuning methods for LLMs.

2. Clear Structure and Writing: The manuscript is well-organized, with a clear presentation of the problem, methodology, experiments, and results.

**Weaknesses:**

Weaknesses:

1. **Marginal improvements for memory**: While the manuscript emphasizes the superior memory efficiency of the LOZO algorithm over MeZO, the proposed advantage is not convincingly demonstrated. The potential improvements in memory usage could be predominantly attributed to MeZO's zeroth-order optimization approach, with LOZO offering only marginal enhancements. As illustrated in Table 1, the memory usage is reduced from a range of approximately 3 to 7.42 GB for MeZO to 2.84 GB for LOZO, which does not present a substantial difference to assert the claimed memory efficiency advantage of LOZO.

2. **Experiments insufficient**: Furthermore, the study's benchmark does not encompass key capabilities of LLMs, such as common sense reasoning (MMLU) and complex reasoning tasks (GSM8k). There is a concern that this proposed fine-tuning approach might not effectively enhance the LLM's high-level cognitive abilities.

**Questions:**

Questions:

1. How do the memory savings of LOZO when applied to larger models and datasets, such as OPT-13B or Llama-70B?

2. Have there been any experiments conducted to assess the impact of this fine-tuning method on the complex capabilities of LLMs, such as instruction following and reasoning?

3. Are there any experiemnts to demonstrate the improvements in  speed of training and convergence when comparing LOZO with current methods?

---

> ### Author Response · Authors · 2024-11-21
>
> - We appreciate your thorough review and valuable feedback on our manuscript. Below, we address the weaknesses you pointed out.
>
> 1. **Clarification of Memory Efficiency**
>
>    Thank you for your detailed feedback. We would like to address a key misunderstanding regarding the focus of our contributions. The primary contribution of our work is not to demonstrate the general memory efficiency of LOZO compared to first-order algorithms, as this aspect has already been established by MeZO. Nor is it to showcase memory efficiency over vanilla MeZO. Instead, our critical contributions are as follows:
>
>    (1) **Performance improvement over vanilla MeZO.** LOZO leverages the low-rank structure in gradient estimation to improve the accuracy performance of MeZO while maintaining nearly the same (and often slightly lower) memory cost. This is evident in the results presented in Tables 1–3, where LOZO significantly outperforms MeZO in terms of accuracy, highlighting its effectiveness.
>
>    (2) **Substantial memory saving compared to momentum-based MeZO.** By exploiting low-rank gradient estimation, we propose LOZO-M, a momentum variant with negligible additional memory overhead for low-rank momentum storage. In contrast, MeZO-M requires storing a full-rank momentum variable, resulting in significantly higher memory consumption. Specifically, as illustrated in **Table 1**, LOZO-M requires only **2.84 GB** of memory, compared to **5.89 GB** for MeZO-M (a nearly **52%** reduction). Given that momentum techniques can substantially improve performance across various tasks (as evidenced in **Tables 8 and 9**), addressing momentum's memory overhead represents a practical and impactful contribution.
>
>    We believe these contributions provide meaningful advancements over MeZO.
>
> 2. **Extension to Additional Reasoning Tasks**
>
>    To evaluate LOZO's effectiveness to enhance LLM's high-level cognitive abilities, we conducted additional evaluations on the WinoGrande dataset which assesses the reasoning abilities. The experiments were conducted using various sizes of the LLaMA model. (Due to limited computational resources,
>    we were unable to test the performance of gradient-based FT on LLaMA-70B.) The results, summarized in the table below, demonstrate that LOZO outperforms MeZO in most scenarios, highlighting its effectiveness in promoting high-level cognitive abilities.
>
>    | Model      | LLaMA-7B | LLaMA-13B | LLaMA-70B |
>    |------------|----------|-----------|-----------|
>    | **LOZO**   | **66.0** | **67.6**  | **72.1**  |
>    | **MeZO**   | 64.3     | 67.2      | **72.1**  |
>    | **FT-LoRA**| 70.9     | 76.6      | 50.4      |
>    | **FT**     | 64.4     | 73.3      | -         |
>
> - Below, we provide our detailed responses to the questions you raised.
>
> 1. **Memory Comparison for OPT-13B and LLaMA-70B**
>
>    We provide a comparison of memory consumption for the **OPT-13B** and **LLaMA-70B** models, as shown below. The MultiRC dataset was evaluated on both OPT-13B and LLaMA-70B. Due to limited computational resources, we were unable to include exact memory cost evaluations for LLaMA-70B using gradient-based full fine-tuning (FT). However, our findings confirm that LOZO-M offers significant memory savings compared to MeZO-M (approximately 50% reduction) and other gradient-based methods, such as FT and FT-LoRA.
>
>    | Optimizer  | OPT-13B (Memory) | LLaMA-70B (Memory) |
>    |------------|------------------|--------------------|
>    | **LOZO**   | 26.9 GB          | 135.5 GB               |
>    | **LOZO-M** | 27.3 GB          | 138.1 GB            |
>    | **MeZO**   | 27.3 GB          | 136.0 GB               |
>    | **MeZO-M** | 52.1 GB          | 270.0 GB                 |
>    | **FT-LoRA**| 102.4 GB         | 187.2 GB                 |
>    | **FT**     | 315.2 GB         | > 640 GB                 |
>
>     We have included these results in **Tables 10 and 12** of the revised manuscript.
>
> 2. **Evaluation on Reasoning and Instruction Following Datasets**
>
>    We have evaluated the WinoGrande dataset on the LLaMA model, and additional tests on several other datasets are presented in Table 11 of the updated manuscript.
>
> 3. **Convergence Speed Comparison**
>
>    In our original manuscript, Figure 3 provided a comparison of the convergence speed between LOZO and MeZO. In the revised version, we have expanded this analysis by adding two additional experiments to evaluate convergence speed across different datasets. Furthermore, we now include a comparison of the GPU wall-clock time for LOZO and MeZO.
>    These new results are presented in Figure 5 in Appendix D.2 of the revised manuscript. These results demonstrate that the LOZO algorithm achieves faster convergence than the MeZO algorithm.
>
> We hope these responses can address your concerns. If you have further questions, we would be happy to provide additional clarification.

---

> > ### Comment · Reviewer_4TZ5 · 2024-11-26
> > **Update**
> >
> > Thank you to the authors for their detailed responses and additional experiments. I appreciate that some of my concerns, such as the wall-clock time comparison in Appendix D.2, have been partially addressed. I am open to raising my score to 6.
> >
> > However, I remain concerned about the fine-tuning performance of the proposed method on complex tasks, particularly on the MMLU benchmark (if GSM8k is tested, it will be better). This dataset is a critical standard for evaluating large models and a meaningful test of their comprehensive capabilities. Since the proposed LOZO method is a parameter-efficient fine-tuning approach for large-scale models, it is essential to validate its performance on challenging tasks to ensure that fine-tuning efficacy is not compromised. Unfortunately, the current experiments do not provide this evidence, and the authors have not directly addressed this concern.
> >
> > If there are any relevant results that I may have overlooked, I would appreciate it if the authors could point them out.

---

> ### Author Response · Authors · 2024-12-02
>
> Thank you for taking the time to review our manuscript and for providing valuable feedback once again.
>
> In response to your concern, we conducted additional experiments on the MMLU benchmark. Due to time constraints, we focused on fine-tuning a single dataset and used the fine-tuned model for testing. The results are presented in the table below:
>
> | Optimizer      | STEM   | Humanities | Social Sciences | Other   | Average |
> |----------------|--------|------------|------------------|---------|---------|
> | **LOZO**       | 37.90  | 44.72      | 56.26            | 55.42   | 48.08   |
> | **MeZO**       | 37.46  | 44.46      | 55.80            | 54.88   | 47.68   |
> | **FT**         | 38.66  | 45.50      | 57.32            | 55.55   | 48.78   |
> | **Original**   | 37.58  | 44.08      | 55.77            | 54.72   | 47.54   |
>
> These results were obtained by fine-tuning the LLaMA-13B model on the WinoGrande dataset using different methods, followed by testing on the MMLU benchmark.
> For comparison, we also include the performance of the original pre-trained model as a baseline. The results clearly demonstrate that LOZO outperforms MeZO, suggesting that LOZO is more effective at learning and retaining information for complex tasks.
>
> We hope these responses can address your concerns. If you have further questions, we would be delighted to address any additional questions or concerns you may have.

---

### Official Review · Reviewer_MNxM · 2024-11-03

**Soundness:** 4
**Presentation:** 4
**Contribution:** 3
**Rating:** 8
**Confidence:** 4

**Summary:**

The authors propose a novel zeroth-order optimization method for fine-tuning. The proposed method consumes significantly less memory while maintaining (and sometime even improving) the quality when compared with other FT methods including MeZO (another zeroth order method), ICL and LoRA methods. The core contribution is the "lazy sampling strategy" where the perturbation matrix for gradient estimation is sampled over several training steps, rather than each iteration. This ensures that the model sufficiently explores low rank sub space, without abrupt changes in the parameters at each iteration step. Extensive Experimentation on large scale OPT models show the efficacy of the approach.

**Strengths:**

1) Well motivated approach, clearly outlines the shortcoming of other zeroth order approaches, like MeZO.
2) Proposed algorithm is well written, clear and the core concepts are presented well. The "lazy sampling strategy" is novel and interesting.
3) The paper proposes momentum variant and provides convergence analysis by interpreting LOZO as subspace optimization method by employing ZO gradient estimator.
4) Extensive experiments on both medium scales and large scale LLMs are convincing in terms of quality gains and memory savings.

**Weaknesses:**

1) The experiments are performed on OPT based LLMs. It would be good to see what kind of memory savings and quality improvements the method gets on SoTA models like Llama.
2) Additionally, the LLM evaluations are not exhaustive and lack eval suites around critical benchmarks around reasoning, MATH, Instruction following etc.
3) An Ablation study on hyper-parameter choices for N and r for critical evals maybe helpful.

**Questions:**

1) Line 268-269. Is there a justification for the chose hyper-parameter values of N and r? Are there ablation results for the same?
2) Figure 3 : Are there loss curves for other datasets at 13B and 30B scale to understand the scaling behavior?

---

> ### Author Response · Authors · 2024-11-21
>
> Thank you for taking the time to review our manuscript. We greatly appreciate your valuable feedback. Below, we provide our responses to your comments:
>
> - Questions:
>
> 1. **Ablation Study**:
>
>    Thank you for the question. We would like to clarify that our original manuscript already includes an ablation study on these hyperparameters, as detailed in Appendix C.3. Based on the results in Figure 4 and Table 7, we observed that a small $r$ (e.g., $r=2$) is sufficient to achieve high accuracy. However, increasing $r$ (e.g., $r=8$) does not lead to performance improvements and can sometimes degrade performance. Given that a larger $r$ introduces additional memory and computational overhead, we limited $r$ to a maximum of 8 in our experiments.
>
>    Furthermore, regarding the hyperparameter $\nu$ (which you referred to as $N$. It appears we do not have a hyperparameter $N$, so we assume you are referring to the subspace period duration $\nu$. Please let us know if you meant a different hyperparameter), our ablation study in Figure 4 and Table 7 shows that a very small $\nu$ negatively impacts convergence. This is likely due to frequent subspace shifts causing abrupt model changes, which destabilize the training process. Conversely, while a larger $\nu$ has a less pronounced effect, it also slightly reduces the algorithm's performance. Based on these findings, we typically set $\nu$ to 50 or 100.
>
> 2. **Additional Curves**:
>
>    We have added two new curves in Figure 5 of Appendix D.2 in the revised manuscript, corresponding to OPT-13B and OPT-30B on two additional datasets. To enable a more comprehensive comparison, we have also included a comparison of wall-clock time on GPUs between our proposed LOZO and the MeZO algorithm.
>
> - Weaknesses:
>
>     Weakness 3 has already been addressed in our response to your second question. Below, we provide response to Weaknesses 1 and 2:
>
>     - Following your comments, we have included experiments on LLaMA models of different model sizes (7B, 13B, and 70B) on various tasks. The memory saving are shown in the following table (see also the newly added Table 12 in the revised manuscript). It is observed that LOZO can save significant memory compared to FT and FT-LoRA.
>
>         | Optimizer   | LLaMA-7B     | LLaMA-70B        |
>         |-------------|--------------|------------------|
>         | LOZO        | 14.1 GB      | 135.5 GB         |
>         | MeZO        | 14.3 GB      | 136.0 GB         |
>         | FT-LoRA     | 32.7 GB      | 187.2 GB         |
>         | FT          | 281.6 GB     | 640 + GB         |
>
>     - The accuracy improvements of LOZO on LLaMA-7B across various tasks are summarized in the following table (The superior results achieved by ZO methods are highlighted in **bold**). Notably, we also evaluated LOZO's performance on the WinoGrande dataset, which assesses reasoning abilities. Due to time constraints, we were unable to test additional MATH and instruction-following tasks. For results on LLaMA-13B and LLaMA-70B, please refer to the newly added Table 11 in the revised manuscript.
>
>
>       | **Task**   | **SST-2** | **WiC** | **COPA** | **SQuAD** | **WinoGrande** |
>       |------------|-----------|---------|----------|-----------|----------------|
>       | **LOZO**   | **94.8**  | **57.2**| 85.0     | **90.3**  | **66.0**       |
>       | **MeZO**   | 91.6      | 56.3    | **86.0** | 90.0      | 64.3           |
>       | **FT-LoRA**| 95.1      | 69.4    | 84.0     | 91.2      | 70.9           |
>       | **FT**     | 94.2        | 72.3      | 83.0     | 90.6        | 64.4           |
>
> We hope these updates and responses adequately address your concerns. If you have further questions or need additional clarifications, we would be happy to provide them.

---

> > ### Author Response · Authors · 2024-12-01
> >
> > Thank you for your time and effort in reviewing our manuscript once again. We have made every effort to address all of your concerns. Could you kindly confirm whether our responses adequately address your questions? If you have any further inquiries, please do not hesitate to contact us and we remain more than willing to provide further explanations as needed.

---

### Official Review · Reviewer_DRFS · 2024-11-06

**Soundness:** 3
**Presentation:** 3
**Contribution:** 3
**Rating:** 6
**Confidence:** 4

**Summary:**

The paper introduces low-rank zeroth-order optimization algorithms, called LOZO and LOZO-M, for memory-efficient fine-tuning of large language models (LLM). The authors claim that by utilizing a low-rank unbiased gradient estimator, LOZO and LOZO-M perform comparably to first-order (FO) methods while outperforming existing zeroth-order (ZO) approaches in term of memory and accuracy. The paper provides convergence guarantees and extensive experimental results to support these claims.

**Strengths:**

The paper proposes a novel algorithm to addresses a critical limitation of ZO methods which is the inability to capture low-rank gradient structures effectively. The application of momentum without substantial memory overhead is innovative. These features are valuable for fine-tuning LLMs in memory-constrained environments. LOZO and LOZO-M achieves comparable performance to traditional FO FFT methods and outperforms existing ZO methods shows that the proposed algorithms can be viable alternatives to widely used FO approaches. The rigorous convergence provides valuable insights into the understanding of LOZO.

**Weaknesses:**

The paper lacks direct memory comparisons with full fine-tuning (FFT) and FT-LoRA methods. This weakens the claim that LOZO and LOZO-M outperforms FO approaches in term of memory efficiency.

Furthermore, the paper primarily focuses on SuperGLUE benchmarks which is limited. Expanding experiments to more tasks would help demonstrate the generalizability of LOZO across different NLP tasks.

Next, one of the main claims of the paper is the ability of LOZO's to handle long-context tasks. However, focusing on SuperGLUE benchmarks only doesn't support this claim.

Finally, testing LOZO and LOZO-M on different types of models, especially, larger models would provide a stronger case for their scalability. Currently, the authors only test the proposed methods on one model family.

**Questions:**

1. Could you provide direct comparisons with FFT and FT-LORA in term of memory usage? This would strengthen your claim of LOZO's memory efficiency relative to FO methods.

2. How do LOZO and LOZO-M perform on tasks beyond SuperGLUE, especially ones tailoring to long-context? This would demonstrate their adaptability and robustness across various applications, and their ability to handle long-context.

3. How do LOZO and LOZO-M perform on different models, especially larger ones? This benchmark would further provide insights into their scalability.

---

> ### Author Response · Authors · 2024-11-21
>
> Thank you for taking the time to review our manuscript. We greatly appreciate your valuable feedback. Below, we provide our responses to your comments:
>
> - Questions:
> 1. **Memory Cost Comparison**:
>
>    We conduct new experiments to compare the memory consumption of the proposed LOZO and LOZO-M algorithms with FFT and FT-LoRA on OPT-13B across two datasets. The results are shown in the following table (see also the newly added Table 10 in the revised manuscript):
>
>    | Optimizer   | RTE (Memory) | MultiRC (Memory) |
>    |-------------|--------------|------------------|
>    | LOZO        | 27.0 GB      | 26.9 GB          |
>    | LOZO-M      | 27.4 GB      | 27.3 GB          |
>    | FT-LoRA     | 79.0 GB      | 102.4 GB         |
>    | FT          | 250.0 GB     | 315.2 GB         |
>
>    It is observed that both LOZO and LOZO-M achieve significant memory savings compared to FT-LoRA and FT. Additionally, we evaluated the memory consumption of LOZO and FT on LLaMA models of varying scales on the MultiRC dataset. The results are provided below (see also the newly added Table 12 in our manuscript).
>
>    | Optimizer   | LLaMA-7B     | LLaMA-70B        |
>    |-------------|--------------|------------------|
>    | LOZO        | 14.1 GB      | 135.5 GB         |
>    | FT-LoRA     | 32.7 GB      | 187.2 GB         |
>    | FT          | 281.6 GB     | 640 + GB         |
>
>    The above results also demonstate the memory efficiency compared to FT and FT-LoRA.
>
> 2. **Experiments Beyond SuperGLUE**:
>
>    Thank you for your comment. We would like to clarify that our original manuscript already includes evaluations of our methods with the OPT model on additional datasets beyond the SuperGLUE benchmark, including SST-2, SQuAD, and DROP (see Tables 2 and 3). These results demonstrate the applicability of our approach to a broader range of tasks. Notably, SQuAD and DROP involve relatively long contexts, further showcasing the robustness of our methods.  In our revised manuscript, we have highlighted the corresponding text in Section 5 to address your concerns.
>
>    To further demonstrate the generalizability of our proposed algorithm across language models, we have conducted additional experiments on LLaMA with datasets beyond SuperGLUE. These results are presented below are for LLaMA-7B (see also the results for LLaMA-13B and LLaMA-70B in the newly-added Table 11 of the revised manuscript).
>
>    | **Task**   | **SST-2** | **WiC** | **COPA** | **SQuAD** | **WinoGrande** |
>    |------------|-----------|---------|----------|-----------|----------------|
>    | **LOZO**   | **94.8**  | **57.2**| 85.0     | **90.3**  | **66.0**       |
>    | **MeZO**   | 91.6      | 56.3    | **86.0** | 90.0      | 64.3           |
>    | **FT-LoRA**| 95.1      | 69.4    | 84.0     | 91.2      | 70.9           |
>    | **FT**     | 94.2        | 72.3      | 83.0     | 90.6         | 64.4           |
>
>    In addition, to evaluate the performance on datasets with long-contexts, we conduct a new experiment on the TREC dataset, which is part of the LongBench benchmark. This experiment evaluated the performance of LOZO against MeZO on RoBERTa-large and OPT-13B. The results are presented in the table below:
>
>    | Optimizer   | RoBERTa-large (TREC) | OPT-13B (TREC) |
>    |-------------|----------------------|----------------|
>    | LOZO        | 77.9                 | 63.2           |
>    | MeZO        | 62.4                 | 25.4           |
>    | FT-LoRA     | 75.8                 | -              |
>    | FT          | 83.7                 | 75.8           |
>
>    Due to the incompatibility between FSDP and LoRA, we were unable to perform FT-LoRA on OPT-13B within the limits of our computational resources.
>
> 3. **Scalability Across Model Sizes on Different Model Families**:
>
>    To evaluate scalability across different model types and sizes, we performed additional experiments on the LLaMA family, including LLaMA-7B, 13B, and 70B. Below, we present the results for LLaMA-7B. For the results on LLaMA-13B and LLaMA-70B, please refer to Table 11 in Appendix D.3 of the revised manuscript. These results demonstrate that LOZO performs well for even larger language models.
>
>    | **Task**   | **SST-2** | **WiC** | **COPA** | **SQuAD** | **WinoGrande** |
>    |------------|-----------|---------|----------|-----------|----------------|
>    | **LOZO**   | **94.8**  | **57.2**| 85.0     | **90.3**  | **66.0**       |
>    | **MeZO**   | 91.6      | 56.3    | **86.0** | 90.0      | 64.3           |
>    | **FT-LoRA**| 95.1      | 69.4    | 84.0     | 91.2      | 70.9           |
>    | **FT**     | 94.2        | 72.3      | 83.0     | 90.6        | 64.4           |
>
> - Weaknesses:
>
>    Regarding the weaknesses you mentioned, we believe they align with the questions raised earlier.
>
> Thank you again for your valuable feedback, and we hope our responses address your concerns. If you have further questions, we would be happy to provide additional clarification.

---

> > ### Comment · Reviewer_DRFS · 2024-11-28
> > **Thanks for the detailed response!**
> >
> > My questions were addressed. I have no further concern.

---

> > > ### Author Response · Authors · 2024-12-01
> > >
> > > Thank you for taking the time to review our manuscript and for providing valuable feedback once again.

---

### Meta-Review · Area_Chair_uTH9 · 2024-12-20

**Metareview:**

The authors propose a low-rank zero-order (ZO) gradient estimator and introduce a novel algorithm, LOZO, which captures the low-rank gradient structure commonly observed in LLM fine-tuning. The proposed method significantly reduces memory consumption while maintaining performance quality compared to other fine-tuning methods, such as MeZO and LoRA. Additionally, the paper introduces a "lazy sampling strategy," wherein the perturbation matrix for gradient estimation is sampled across multiple training steps rather than at every iteration. This approach enables the model to effectively explore the low-rank subspace without abrupt parameter changes at each iteration. The experimental results further demonstrate the efficacy of the approach.

Overall, the reviewers unanimously acknowledge the soundness and contributions of the proposed techniques, and their efficiency improvements. Given the potential impact of reducing the memory costs associated with training large models, we recommend accepting this submission. But for authors, please follow the reviewers' feedback and your promise to revise the paper.

**Additional Comments On Reviewer Discussion:**

I mainly list the key concerns.

1)	Clarification of Memory Efficiency (Reviewer DRFS, 4TZ5).
The authors have clearly discussed existing results to show the memory efficiency and provided extra experimental results for further comparison, addressing the concern about efficiency.

2)	Experiments insufficient (Reviewer MNxM, 4TZ5).
The authors have provided additional experimental results on other reasoning tasks by testing several big models.

3)	Improvement in experiments is limited. （Reviewer FzN7）
The authors have discussed existing results to show performance improvement, which addresses efficiency concerns.

All these key concerns are addressed.

---

### Decision · Program_Chairs · 2025-01-22

Accept (Poster)